# Value Aggregation with Uncertainty in Online Decentralized MARL

**Ziyue Chu** [1]   **Leonardo Stella** [1]

## Abstract

Multi-agent reinforcement learning (MARL) has received increasing attention for solving complex decision-making tasks. Networked MARL approaches offer a decentralized solution for parameter sharing to accelerate training via value aggregation. However, existing federated aggregations rely on convex averaging that may fail to converge to global optima and cause learning rollback in the online learning setting. In this paper, we formally characterize the learning rollback phenomenon arising from aggregating value estimates with unequal uncertainty under heterogeneous online update depths. We propose a novel adaptive global consensus (AGC) mechanism for Q-value aggregation in decentralized MARL policy evaluation, which dynamically adjusts aggregation weights based on agents' uncertainty. We establish theoretical guarantees on accelerated convergence and bounded learning variance with empirical validations, advancing the state-of-art MARL theory.

## 1. Introduction

Multi-agent reinforcement learning (MARL) has received increasing attention for solving complex real-world decision-making tasks in various domains, from smart transportation systems to distributed robotics and quantitative trading. Yet, open problems still remain, including convergence guarantees, sample efficiency and scalability issues (Zhang et al., 2025; Zhang & Stella, 2026; Kim & Sung, 2023). Decentralized MARL is a standard framework to improve efficiency by restricting the state-action space from exponential to linear. However, this brings new challenges that the learning bypasses joint coordination and still suffers from scalability.

Recent research on networked MARL (Chu et al., 2025;

[1]School of Computer Science, College of Engineering and Physical Sciences, University of Birmingham, B15 2TT, Birmingham, United Kingdom. Correspondence to: Ziyue Chu <zxc332@student.bham.ac.uk>.

*Proceedings of the 43ʳᵈ International Conference on Machine Learning*, Seoul, South Korea. PMLR 306, 2026. Copyright 2026 by the author(s).

Zhang et al., 2018a) demonstrates a promising avenue to accelerate learning and further reduce sample complexity by utilizing exchanged information among agents during learning. However, direct experience data sharing can compromise the overall data privacy of the system, and requires large memory buffer and communication overheads. Therefore, many networked MARL algorithms resort to parameter sharing as a communication-efficient and privacy-preserving alternative (Gupta et al., 2017; Tan, 1993). This produces consensus on parameters after aggregation.

Federated-style parameter aggregation adopts convex weights in forming a consensus parameter, which gains huge popularity in eliminating uncertainty for uniformly i.i.d sampled data (Konečný et al., 2016; McMahan et al., 2017). The success of such federated learning lies in convex aggregation over random variables with the same expectation reducing variance. Fed-Q learning applies similar techniques to generate the consensus Q-function by aggregating Q-values at each state-action among all agents in dec-MARL (Zhuo et al., 2019). However, the constraint on uniformly i.i.d. sampled data can be hardly satisfied in online multi-agent learning (Li et al., 2020b; Jin et al., 2022). Each agent collects online data from different regions of the state space upon initialization, and the learned Q-values get heterogeneous uncertainty with various update depth. This violation results in different bias-level merging and causes learning rollback and convergence failure.

Motivated by the above, our **research question** is:

> *How can we design an efficient online algorithm to bypass learning rollback for uncertain value aggregation in networked dec-MARL?*

After formally characterize the learning rollback phenomenon, our contribution is threefold:

(1) A novel adaptive global consensus (AGC) mechanism for agents parameter aggregation with different value uncertainty, favoring more informative updates.

(2) A novel DTDE learning dynamics that recursively updates the adaptive consensus weights in (1).

(3) Theoretical learning guarantees for faster convergence and bounded variance, with empirical validation.

## 1.1. Significance Analysis on Learning Rollback

As a widely adopted approach for parameter consensus in networked decentralized MARL, existing federated-style learning methods commonly aggregate parameters across agents using convex weights, such as FedAvg (Li et al., 2019). These approaches are theoretically effective when agents' estimates are sampled from approximately homogeneous or uniformly explored state-action distributions, i.e., under the **uniform ergodic assumption** (Woo et al., 2025), on which most existing theoretical analyses are built. However, such assumptions rarely hold in realistic online learning settings. In practice, agents continuously interact with the environment under heterogeneous trajectories and unequal visitation frequencies, resulting in heterogeneous uncertainty (both bias and variance) over the state-action space **due to non-uniform visitation**. This is a fundamental property of online learning (Albrecht et al., 2024).

The learning rollback phenomenon has also been observed in parallel real-world experiments on online cooperative learning in decentralized MARL systems (Chu et al., 2025; Huanca et al., 2026), where convex aggregation over parameters with different uncertainty levels degrades the estimates of more confident agents, thereby slowing or even reversing the overall learning progress. This issue becomes increasingly pronounced in **large-scale** or **sparse state-action spaces**, where local estimates remain highly biased in many regions due to limited visitation. In such settings, there is higher chance for convex aggregation to increase the uncertainty of confident estimates, despite only marginal reduction of the poorly trained agents.

From a broader perspective, this paper also addresses the question: "Are More Heads better than Fewer but Smarter?" when considering decentralized learning strategies in practical multi-agent scenarios. Our answer is not always yes, since aggregated parameters do not necessarily form an effective consensus and may instead be hindered by noisy or highly uncertain estimates. Starting from, but not limited to, the tabular setting, this paper establishes a framework that bridges aggregation weight design and estimation uncertainty to achieve uniform convergence guarantees in networked MARL with parameter sharing scheme when uncertainty can be quantified.

## 1.2. Paper Organization

The remainder of this paper is organized as follows. Section 2 introduces the background of online networked decentralized MARL and reviews federated-style parameter aggregation methods. Section 3 formally characterizes the learning rollback phenomenon arising from aggregating value estimates with heterogeneous uncertainty under non-uniform online updates, and explains why traditional convex aggregation in federated-style learning fails in online learning settings. Section 4 presents the proposed Adaptive Global Consensus (AGC) framework and the decentralized recursive algorithm for uncertainty-aware value aggregation. Section 5 provides the theoretical analysis of the proposed method, including the properties of the AGC weights, decentralized computability, convergence acceleration, and bounded variance guarantees. Section 6 empirically validates the proposed framework through symbolic MDP and cooperative multi-agent navigation experiments, demonstrating the existence of learning rollback and the effectiveness, scalability, and robustness of AGC. Section 7 reviews related work on networked decentralized MARL. Finally, Section 8 concludes the paper and discusses future research directions.

## 2. Background & Preliminaries

A multi-agent control problem can be formulated as a Markov game: $\mathcal{M} = ([M], \mathcal{S}, \{\mathcal{A}_i\}_{i=1}^M, \mathbb{P}, \{R_h\}_{h=1}^H, \gamma)$, which can be decomposed into a series of parallel MDPs if agents are homogeneous (defined in Assumption 2.1):

$$\left\{ (\mathcal{S}_i, \mathcal{A}_i, \{\mathbb{P}_h\}_{h=1}^H, R_{h,i}, \gamma) \right\}_{i=1}^M, \tag{1}$$

where each agent in the agent set $[M]$ shares the same finite discrete state-action space $\mathcal{S}_i = \mathcal{S}_j, \mathcal{A}_i = \mathcal{A}_j, \forall i, j \in [M]$, and the joint state-action space becomes the Cartesian product of local spaces, i.e., $\mathcal{S} = \bigotimes_{i=1}^M \mathcal{S}_i$.

The system transition kernel defines the system dynamics and factorizes as product of individual $\mathbb{P} : \mathcal{S} \times \{\mathcal{A}_i\}_{i=1}^M \mapsto \mathcal{S}$, $\mathbb{P}(s_h, a_h) = \prod_{i=1}^M \mathbb{P}_h(s_{h,i}, a_{h,i})$, and the reward function decomposes $R_h = \sum_i R_{h,i}$ at timestep $h$. The discount factor $\gamma \in (0, 1)$ controls the weight of future rewards.

**Assumption 2.1** (Homogeneity). All agents in the MAS share the same learning goal and exhibit symmetric policy effects, such that the permutation of agent indices does not alter the system's joint policy evaluation $V$:

$$V_i^{\boldsymbol{\pi}} = V_{\sigma(i)}^{\langle \pi_{\sigma(1)}, \pi_{\sigma(2)}, \dots, \pi_{\sigma(M)} \rangle}, \ \forall i \in [M], \tag{2}$$

where joint policy $\boldsymbol{\pi} = (\pi_1, \pi_2, \dots, \pi_M)$ and $\sigma(\cdot)$ denotes a permutation mapping on $[M]$.

Assumption 2.1 is a widely used assumption in both Dec-MARL theory (Albrecht et al., 2024) and practice (i.e. swarm and mean-field games (Hüttenrauch et al., 2019; Guo et al., 2019)). This decomposition simplifies the joint policy analysis and computation in a $M$-linear state space, leading to the following proposition.

**Proposition 2.2.** *The optimal joint policy $\boldsymbol{\pi}^*$ in the MA-MDP under Assumption 2.1 consists of identical individual optimal policies: $\pi_i^* = \pi_j^*, \forall i, j \in [M]$.*

## 2.1. Online Networked MARL

Networked MARL has shown great power to accelerate learning for finding individual optima with information exchange networks (Chu et al., 2025; Zhang et al., 2018a). In the online decentralized training and decentralized execution (DTDE) setting, each agent learns locally while exchanging information with its neighbors through a network.

In detail, each agent $i$ observes its local state $s_{h,i}$, selects an action $a_{h,i} \in \mathcal{A}_i$ according to its policy $\pi_{h,i}(a|s_{h,i})$, receives a reward $r_{h,i} \sim R_{h,i}$, and transitions to the next state $s_{h+1,i} \sim \mathbb{P}_h$. Given an initial state $s_{1,i}$, the trajectory under policy $\pi_{h,i}$ is $\{(s_{h,i}, a_{h,i}, s_{h+1,i}, r_{h,i})\}_{h=1}^{H}$. The objective of each agent is to maximize its policy evaluation $V_{1,i}^{\pi_i}$:

$$\pi_i^* = \arg\max_{\pi_i} V_{1,i}^{\pi_i}, \quad V_{1,i}^{\pi_i} = \mathbb{E}\left[\sum_{h=1}^{H} \gamma^h r_{h,i}\right]. \quad (3)$$

Similarly, we define the state-action function for Q-values that takes one specific action different from $\pi_i$ in $V_{1,i}^{\pi_i}$. The policy evaluation is estimated through contraction steps from the initialized value to the cumulated reward, where fast evaluation is the key to algorithm efficiency. For infinite-horizon MDPs, networked MARL divides the online learning process into $K$ episodes, each with a finite horizon $H$.

On top of local data collection and updates, online networked MARL introduces communication net to exchange information (e.g. learned parameters or value estimates) among agents (Zhang et al., 2018b; Gronauer & Diepold, 2022). The additional shared information promotes individual learning convergence to improve sample efficiency. The info-sharing structure can be formulated as an undirected graph $\mathcal{G} = ([M], \mathcal{E})$, where $[M]$ is the set of vertices (agents) and $\mathcal{E}$ the set of edges. Its adjacency matrix $A \in \{0,1\}^{M \times M}$ is defined with elements

$$a_{ij} = \begin{cases} 1, & \text{if } (i,j) \in \mathcal{E}, \\ 0, & \text{otherwise.} \end{cases}$$

To simplify later theoretical analysis, we adopt a fully connected network with $A = \mathbf{1}\mathbf{1}^\top$. Appendix A (Algorithm 2) summarizes the detailed online DTDE learning framework.

### 2.2. Parameter Sharing with Federated Aggregation

Considering information exchange privacy, sharing local parameters (i.e., Q-values, NN parameters) as abstracted information is a popular approach in networked MARL (Christianos et al., 2021; Wang et al., 2024). The shared parameters then form a global consensus among agents in the global parameter update step. A standard method to make such global consensus is via federated aggregation (Zhuo et al., 2019) with convex averaging over all shared parameters.

Let the local value estimations $\hat{Q}_{t,i}$ at time $t$ for any fixed state-action pair be the target aggregation parameter, the one step global consensus making updates the value to

$$\hat{Q}_{t',m} \leftarrow \sum_{i \in [M]} w_{i,m} \hat{Q}_{t,i}, \forall m \in [M],$$
$$\text{s.t.} \quad \sum_{i \in [M]} w_{i,m} = 1; w_{i,m} \geq 0, \forall i \in [M], \quad (4)$$

where $w_{i,m} \in [0,1]$ are the weights in the convex parameter space reflecting how much the observed $\hat{Q}_{t,i}$ from agent $i$ affects agent $m$ in obtaining the consensus value.

**Lemma 2.3.** *If value estimates $\hat{Q}_{t,i}$ uniformly converge to the optimal Q-function, i.e. $|\hat{Q}_{t,i} - Q^*| < \epsilon, \forall t > T(\epsilon)$, $i \in [M]$, the convex consensus in equation (4) also uniformly converges to the same optima and we indicate that as: $\hat{Q}_{t',m} \rightrightarrows Q^*$ with $t' = t$.*

## 3. Learning Rollback Formulation

Before formally defining the learning rollback issue, we provide a vector representation in Fig. 1 to illustrate how the issue occurs in traditional federated methods.

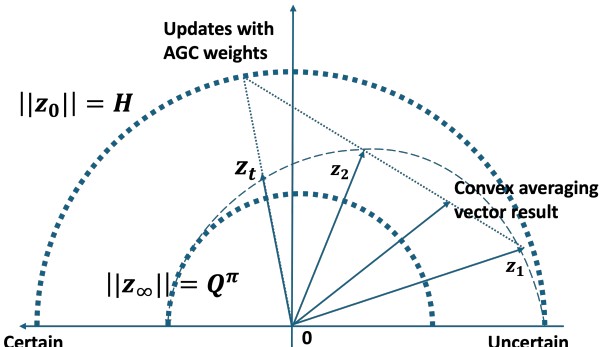

*Figure 1.* Vectorized representation of the learning rollback.

Here, the coordinate system is a **polar plane**, where the polar angle and vector norm are the quantities of interest. Vectors represent learned parameters (e.g., $\mathbf{z_1}$, $\mathbf{z_2}$ for the estimated parameters of two agents), and the thin dashed lines represent the learning trajectories connecting the end points of a vector. The polar angle between the vector and the negative $x$-axis measures the confidence level of a learned parameter within the range $[0°, 180°]$, $0°$ for the negative $x$-axis (estimation is uncertain) and $180°$ for the positive $x$-axis. Two reference semi-circles (inner and outer) represent the values for the true policy evaluation $Q^\pi$ and its maximal (optimistic) initialization $||\mathbf{z_0}||$, respectively.

The goal of the global consensus learning is to reach the converged vector length $Q^\pi$ (inner semicircle) while reducing uncertainty (increasing vector polar angle) for all agents,

i.e., $\mathbf{z_t}$ as an ideal learning trajectory. However, when the agents reach consensus through convex averaging, the output vector lies between $\mathbf{z_1}$ and $\mathbf{z_2}$, always slowing down the training process for the well-trained (more confident) agent $\mathbf{z_2}$. This phenomenon is called *learning rollback*.

Theoretically, the convex weights constraint in equation (4) only ensure that the aggregated Q-function converges to the optima under the uniform convergence assumption of each agent as in Lemma 2.3. However, this uniform convergence is hard to achieve in general, especially for online learning with heterogeneous data, where online Q-learning algorithms can only get point-wise convergence, namely:

$$|(\hat{Q}_{t,i} - Q^*)(s,a)| < \epsilon_i, \forall t > T(\epsilon_i, s, a), \ i \in [M]. \quad (5)$$

Here, $T$ is a function that also depends on the inputs $(s, a)$.

In online learning, the total number of timestep $t$ updates are distributed in the state-action space, such that

$$\sum_{s,a} t_i(s,a) = t, \forall i \in [M], \quad (6)$$

where $t_i(s, a)$ is agent $i$'s actual number of updates on $(s, a)$.

Note that $t_i(s, a)$ is irrelevant w.r.t. timestep $t$ in general, i.e., $t_i(s, a) = 0$ as training steps $t$ increases in the worst case, and $\epsilon_i = T^{-1}(t_i(s, a))$ is only relevant with the actual $t_i$. If we want a uniform convergence result as Lemma 2.3, we can fix input $(s, a)$, and the convex aggregation can only guarantee the following gap with an input-irrelevant $T(\epsilon)$, provided that the gap $\max_i\{\epsilon_i\}$ cannot be arbitrarily small:

$$|(\hat{Q}_{t',m} - Q^*)(s,a)| < \max_i\{\epsilon_i\}, \forall t > T(\epsilon), \ i \in [M]. \quad (7)$$

Conversely, under a uniform distribution $t_i(s,a) = t/(|S||A|)$, the gap $\max_i\{\epsilon_i\} = \epsilon_i, \forall i \in [M]$ can be arbitrarily small with $\epsilon$ to keep uniform convergence. This mismatch between $\max_i\{\epsilon_i\}$ and $\epsilon$ is caused by heterogeneous exploration and nonuniform local updates.

With additional constraints in the parameter search, the following formulation can guarantee uniform convergence:

$$\hat{Q}_{t',m} \leftarrow \sum_{i \in [M]} w_{i,m} \hat{Q}_{t,i}, \forall m \in [M],$$

$$\text{s.t.} \quad \sum_{i \in [M]} w_{i,m} = 1; w_{i,m} \geq 0, \forall i \in [M],$$

$$||\hat{Q}_{t',m} - Q^*||_\infty < ||\hat{Q}_{t,i} - Q^*||_\infty, \forall i, m \in [M]. \quad (8)$$

*Proof.* The proof is given in Appendix D.2. $\qquad\square$

**Proposition 3.1.** *Equation* (8) *has no solution in the convex weight space.*

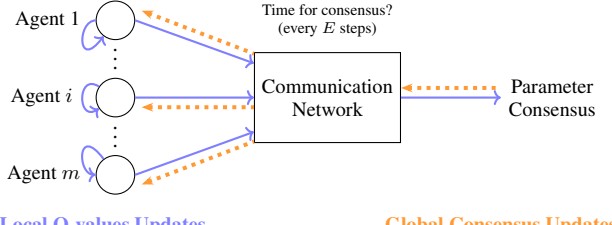

*Figure 2.* Diagram showing local and global updates.

However, we can relax the convex constraints for finding global consensus weights and show the existence of the solution and its learnability, which is the focus of this paper.

## 4. Our Approach – Adaptive Global Consensus

To overcome the issue of learning rollback with the uniform convergence constraint in equation (8), we first formulate our Adaptive Global Consensus (AGC) weights in value aggregation for dec-MARL, as a solution for equation (8). The whole learning process includes local Q-value updates and global consensus updates. Local Q-value updates execute online Q-evaluation for policy iteration on each terminal agent in parallel, while global consensus updates are executed every $E$ steps to learn the designed AGC in a decentralized manner as shown in Figure 2. The learned consensus is then transferred back to each agent through the communication network to overwrite each agent's Q-value belief for later local updates. A detailed workflow for Algorithm 2 is provided in Appendix A.

### 4.1. Local Q-value Evaluation for Policy Iteration

Each agent $m$ optimizes its policy $\pi_{i,m}$ by maximizing its Q-function evaluation $Q^{\pi_{i,m}}$, and the policy is improved by selecting the greedy action according to its Q-evaluation:

$$\pi_{i+1,m}(s) \leftarrow \arg\max_{a'} Q^{\pi_{i,m}}(s, a'). \quad (9)$$

To obtain an accurate Q-evaluation, each agent gathers its real-time private transition data $(s, a, r, s')$ from the environment and updates $Q^{\pi_{i,m}}(s, a)$ independently. Specifically, the local one-step policy evaluation update is:

$$Q^{\pi_{i,m}}_{t+1}(s, a) \leftarrow r(s, a) + \gamma Q^{\pi_{i,m}}_t(s', \pi_{i,m}(s')). \quad (10)$$

Initialized at any value, iterations of equation (10) converge to the true policy evaluation $Q^{\pi_{i,m}}$ because of Bellman's contraction property. The policy evaluation learning curve and its analytical form are summarized in Lemma 4.1.

**Lemma 4.1.** *The iterations in equation (10) satisfies $\gamma$-contraction property under linear operator $\mathcal{T}^{\pi_{i,m}}$:*

$$||(Q^{\pi_{i,m}}_t - Q^{\pi_{i,m}})(s, a)||_\infty \leq \gamma^t \frac{R_{\max}}{1 - \gamma}, \ \forall (s, a) \in \mathcal{S} \times \mathcal{A},$$

where $\|\cdot\|_\infty$ denotes the matrix infinity norm, and $t$ is the actual update number at $(s,a)$. More precisely, the iterated Q-value $Q_t^{\pi_{i,m}}$ decays exponentially from its initialization and converges to the true policy evaluation $Q^{\pi_{i,m}}$:

$$Q_t^{\pi_{i,m}}(s,a) = (1-\gamma^t)Q^{\pi_{i,m}}(s,a) + \gamma^t Q_0(s,a) + M_t, \tag{11}$$

where $M_t$ is a martingale-difference sequence representing unbiased noise (zero mean conditioned on the past) from stochastic sampling. Note that the convergence of equation (11) is only point-wise depending on input $(s,a)$.

After policy evaluation converges, the policy improvement step updates the one-step greedy policy $\pi_{i,m}$, using $Q_t^{\pi_{i,m}}$ as a proxy for $Q^{\pi_{i,m}}$ when $t$ is sufficiently large:

$$\pi_{i+1,m}(s) \leftarrow \arg\max_{a'} Q_t^{\pi_{i,m}}(s,a'). \tag{12}$$

The algorithm terminates when the policy no longer changes (i.e., $\pi_{i+1,m} = \pi_{i,m}$), yielding the optimal policy $\pi^*$ and the corresponding optimal Q-function $Q^{\pi_m^*} = Q_m^*$.

## 4.2. Adaptive Global Consensus Learning

For each target agent $m \in [M]$, denote its neighbor set $\mathcal{N}_m$, Q-table matrix $\hat{Q}_m$, visitation matrix $N_m^t$ with training timestep $t$, consensus-making steps $E$, and the visitation difference matrix between two consensus points $D_m^t = N_m^t - N_m^{t-E}$. Let the element-wise matrix product and power operator be denoted by $\odot$ and $\mathrm{Pow}(A;\gamma) = [\gamma^{a_{ij}}]$, with discount factor $\gamma$ and integer matrix $A = [a_{ij}]_{\mathcal{S}\times\mathcal{A}}$.

For any $(s,a) \in \mathcal{S} \times \mathcal{A}$, if local updates are nonuniform $D_m^t \neq D_i^t, \forall i \in \mathcal{N}_m$, the adaptive global consensus (AGC) weight matrix for agent $m$ and its neighbors $i \in \mathcal{N}_m$ are:

$$\begin{cases} W_m^t = \sum_{i\in\mathcal{N}_m} \dfrac{\mathrm{Pow}(D_i^t) - \mathrm{Pow}(D_m^t + \sum_j D_j^t)}{A_m(\mathrm{Pow}(D_i^t) - \mathrm{Pow}(D_m^t))} \\ W_{i,m}^t = \dfrac{\mathrm{Pow}(D_m^t + \sum_j D_j^t) - \mathrm{Pow}(D_m^t)}{A_m(\mathrm{Pow}(D_i^t) - \mathrm{Pow}(D_m^t))}, \end{cases} \tag{13}$$

where matrix $A_m = \sum_{i\in\mathcal{N}_m} \mathbb{I}[D_m^t \neq D_i^t]$ counts agent number with nonuniform updates at each $(s,a)$.

Here $D_m^t + \sum_{j\in\mathcal{N}_m} D_j^t$ represents the total number of local updates $m$ can get from neighbors, and $\mathrm{Pow}(D)$ measures the uncertainty related to its local updates.

Remark 4.2 (Example of extreme weights for 2 agents).

1. Agent $M$ communicates with agent $N$ with no local updates: $D_n^t = 0, D_m^t \neq 0$; then $W_m^t = 1, W_{n,m}^t = 0$. $M$ retains its estimate, while $N$ adopts $M$'s value.

2. $M$ with a well-trained agent $N$: $D_n^t \to \infty$; then $W_m^t = 0$, $W_{n,m}^t = 1$. Agent $M$ overwrites its value with $N$'s.

3. $M$ with a less confident agent $N$: $D_m^t > D_n^t; W_m^t > 1$, $W_{n,m}^t < 0$, extending the consensus towards the more confident agent $M$ while preserving $W_m^t + W_{n,m}^t = 1$.

**Theorem 4.3.** *AGC weights in equation* (13) *is a solution for equation* (8) *that guarantee uniform convergence.*

*Proof.* The proof is left in the analysis of Section 5. $\qquad\square$

In the case that $D_m^t = D_k^t$ for some $(s,a)$ and $k \in \mathcal{N}_m$, the shared Q-values are assumed to lie within the same uncertainty (error bound) and the learning rollback issue is not happening. We apply traditional simple averaging among all those agents to reduce stochastic noise, which is equivalent to assign equal weights to all such agent $k$ at $(s,a)$ that $D_m^t(s,a) = D_k^t(s,a)$:

$$W_m^t(s,a) = W_{k,m}^t(s,a) = \frac{1}{|\mathcal{N}_m| + 1 - A_m}. \tag{14}$$

The global consensus update for value aggregation with uncertainty is the weighed-average of all received parameters:

$$\hat{Q}_m \leftarrow W_m^t \odot \hat{Q}_m + \sum_{i\in\mathcal{N}_m} W_{i,m}^t \odot \hat{Q}_i, \ \forall m \in [M]. \tag{15}$$

However, in decentralized learning, system-level data, such as $A_m$ and $\sum_{j\in\mathcal{N}_m} D_j^t$, is not available for target agent $m$ when updating its global consensus. To learn the weights in a fully decentralized (DTDE) manner, we provide the following Algorithm 1 to compute the AGC weights.

---

**Algorithm 1** Decentralized AGC Weights Matrix Learning

---

**Require:** Any agent $m$, neighbor set $\mathcal{N}_m$, masks $= [\cdot]$
**Ensure:** $W_{i,m}^t, W_m^t$, Updated $\hat{Q}_m$
1: $D_m^{sum} = \mathbf{0}, A_m = \mathbf{0}$, Parameters memory $\mathcal{P} = [[\cdot]]$
2: **for** each agent $i \in \mathcal{N}_m$ **do**
3: $\quad D_i^{dif} = D_i^t - D_m^t$, $\mathrm{mask}_i = \mathbb{I}[D_i^{dif} = 0]$
4: $\quad \mathrm{masks} \leftarrow \mathrm{masks} + \mathrm{mask}_i, W_{i,m}^t[\mathrm{mask}_i] = 1$
5: $\quad W_{i,m}^t[1 - \mathrm{mask}_i] \leftarrow 1/(\mathrm{Pow}(D_i^{dif}) - 1)$
6: $\quad \mathcal{P} \leftarrow \mathcal{P} + [W_{i,m}^t, \hat{Q}_i], A_m \leftarrow A_m + (1 - \mathrm{mask}_i)$
7: $\quad D_m^{sum} \leftarrow D_m^{sum} + D_i^t \odot (1 - \mathrm{mask}_i)$
8: **end for**
9: Denote $\mathcal{P}_0 = [W_{i,m}^t], \mathcal{P}_1 = [\hat{Q}_i]$ the 2 columns of $\mathcal{P}$
10: **if** $\mathrm{masks}(i,s,a) \neq 1, \forall i,s,a \in \mathcal{N}_m, \mathcal{S}, \mathcal{A}$ **then**
11: $\quad [W_{i,m}^t] \leftarrow \mathcal{P}_0 \odot [(\mathrm{Pow}(D_m^{sum}) - 1) \odot A_m]$
12: **else** $[W_{i,m}^t] \leftarrow 1/(|\mathcal{N}_m| + 1 - A_m)$
13: **end if**
14: $W_m^t \leftarrow 1 - \sum_{i\in\mathcal{N}_m} W_{i,m}^t$
15: $\hat{Q}_m \leftarrow W_m^t \odot \hat{Q}_m + \langle [W_{i,m}^t], \mathcal{P}_1\rangle$

---

The key idea is to compute the denominator of each neighbor's weight $W_{i,m}^t$ first whenever receiving connected messages, while preserving the individual data for the accumulator in the numerator. The numerator is an agent-invariant matrix and can be obtained when all information is exchanged. Target agent's weight matrix $W_m^t = 1 - \sum_i W_{i,m}^t$ can be easily computed via Lemma 5.1.

*Remark* 4.4 (Highlights of Algorithm 1).

- Communication-efficient: lower exchange frequency (every $E$ steps).

- Fully decentralized: recursive weight computation enables DTDE implementation w/o central coordination.

- Accelerated convergence: variance reduction and adaptive weighting promote faster policy evaluation.

## 5. Learning Performance Analysis

This section will first show Algorithm 1 learns our AGC weights matrix. We then analyze the effectiveness of AGC weights in value aggregation with uncertainty.

### 5.1. Properties of AGC Weights Matrix

Before the start of analysis, we first provide the following properties of the weights matrix in Equation (13):

**Lemma 5.1** (Normalization). *The sum of AGC weights matrix in equation* (13) *are normalized:*

$$\sum_{i \in \mathcal{N}_m} W_{i,m}^t + W_m^t = \mathbf{1}, \quad \forall m \in [M].$$

**Lemma 5.2** (Range of weights in equation (13)). *Given any fixed $m \in [M]$, $\forall i \in \mathcal{N}_m, (s,a) \in \mathcal{S} \times A$, the weights in AGC matrix* (13) *are bounded:* $\frac{\gamma^{ME}-1}{1-\gamma} \leq W_{i,m}^t(s,a) \leq \frac{1-\gamma^{ME}}{1-\gamma}$, *and* $\frac{\gamma^{ME}-\gamma}{1-\gamma} \leq W_m^t(s,a) \leq \frac{2-\gamma^{ME}-\gamma}{1-\gamma}$.

*Proof.* Here, we denote $\max |W_{i,m}^t(s,a)| = \frac{1-\gamma^{ME}}{1-\gamma} = Z$ for later use. See Appendix D.3 for detailed proofs. □

### 5.2. Decentralized Computation of AGC Weights Matrix

Lemma 5.3 shows that Algorithm 1 recursively computes the proposed AGC weights matrix, exactly realizing the uncertainty-aware aggregation and reducing to simple averaging for equal uncertainty, as defined in (13)–(14).

**Lemma 5.3.** *The recursively calculated AGC weights matrix for target agent $m$ in Algorithm 1 is equivalent to the designed weight matrix in equations* (13)-(14).

*Proof.* See Appendix D.4. □

Furthermore, global consensus updates in Algorithm 1 computes the weighed-average of all shared parameters from terminal agents of the target agent $m$ following equation (15).

### 5.3. Learning Parameters Estimation

Recall the analytical expression of learning iteration for policy evaluation in equation (11): $Q_t^{\pi_{i,m}}(s,a) = \mu_m +$

$\alpha_m \gamma^t, \forall m \in [M], (s,a) \in \mathcal{S} \times \mathcal{A}$, where $\mu_m$ and $\alpha_m$ are unknown random variables with $\mathbb{E}[\mu_m] = Q^{\pi_{i,m}}(s,a)$ and $\mathbb{E}[\alpha_m] = Q_0(s,a) - Q^{\pi_{i,m}}(s,a)$ are to estimate.

**Theorem 5.4** (Random variables estimation with Pair-wise value aggregation). *At global consensus update, $\forall i \in \mathcal{N}_m, (s,a) \in \mathcal{S} \times \mathcal{A}$ with $D_i^t(s,a) \neq D_m^t(s,a)$:*

$$\begin{cases} \hat{\mu}_{m,i} = \dfrac{(\mathrm{Pow}(D_i^t)\hat{Q}_m - \mathrm{Pow}(D_m^t)\hat{Q}_i)(s,a)}{\mathrm{Pow}(D_i^t)(s,a) - \mathrm{Pow}(D_m^t)(s,a)}, \\[2ex] \hat{\alpha}_{m,i} = \dfrac{\hat{Q}_m(s,a) - \hat{Q}_i(s,a)}{\mathrm{Pow}(N_m^t)(\mathrm{Pow}(D_m^t) - \mathrm{Pow}(D_i^t))(s,a)} \end{cases}$$

*are unbiased estimators for $\mathbb{E}[\mu_m]$ and $\mathbb{E}[\alpha_m]$ respectively given parameters from target-terminal agents pair $m$ and $i$.*

*Proof.* See Appendix D.5. □

**Corollary 5.5.** *Simple average over all pair-wise terminal-target estimations $\bar{\mu}_m = \frac{1}{A_m} \sum_{n_i} \hat{\mu}_{m,n_i}$ and $\bar{\alpha}_m = \frac{1}{A_m} \sum_{n_i} \hat{\alpha}_{m,n_i}$ are also unbiased estimators with reduced variance for $\mathbb{E}[\mu_m]$ and $\mathbb{E}[\alpha_m]$ respectively.*

### 5.4. Faster Convergence and Bounded Variance

Finally, we show the aggregated value after global consensus update in Algorithm 1 accelerates the policy evaluation learning and converges to the true policy evaluation:

**Theorem 5.6** (learning convergence and acceleration). *After one step of global consensus update, the learned parameters have the following property: $\forall (s,a) \in \mathcal{S} \times \mathcal{A}$, $m \in [M]$,*

$$\mathbb{E}[\hat{Q}_m(s,a)] = \mathbb{E}[\mu_m] + \mathbb{E}[\alpha_m] \mathrm{Pow}\left(D_m^t + \sum_{i \in \mathcal{N}_m} D_i^t\right)(s,a),$$

*with bounded variance* $\mathrm{Var}[M_t]/M \leq \mathrm{Var}[\hat{Q}_m(s,a)] \leq ((Z+1)^2 + \frac{Z^2}{M-1})\mathrm{Var}[M_t]$.

*Proof.* The full proof appears in Appendix D.6; only a proof sketch is given here. Replace the quantities $\mathbb{E}[\mu_m], \mathbb{E}[\alpha_m]$ in the result with $\mathbb{E}[\bar{\mu}_m], \mathbb{E}[\bar{\alpha}_m]$ according to Corollary 5.5 and Theorem 5.4. By recalling the definition of the AGC weights matrix in equation (13) and the value aggregation in equation (15), we get the convergence proved. Note that this expectation holds for any agent $m$ in the system, so after value aggregation with the AGC weights matrix, all agents' $Q$-estimations are adjusted to the same error range $\mathrm{Pow}(N_m^{t+E})(s,a) = \mathrm{Pow}(N_m^t + D_m^t + \sum_i D_i^t)(s,a)$.

Next, let us consider the variance of globally updated $\hat{Q}_m$:

$$\frac{\mathrm{Var}[\hat{Q}_m(s,a)]_{new}}{\mathrm{Var}[M_t]} = \frac{W_m^t(s,a)^2}{|\mathcal{N}_m| - A_m + 1} + \sum_i W_{i,m}^t(s,a)^2.$$

We can apply Cauchy–Schwarz inequality and normalization constraints to get its lower bound, while considering the weights range in Lemma 5.2 for the upper bound. □

This theorem establishes that the global consensus update is equivalent to augmenting the target agent $m$ with additional local updates from its connected terminal agents, while $\mathbb{E}[\hat{Q}_m]$ still converges to the true policy value $Q^{\pi_{i,m}}$ as the number of iterations increases. This means that the AGC updates satisfy the uniform convergence conditions in equation (8). Although the algorithm's output $\hat{Q}_m$ has a slightly larger variance compared to traditional convex federated weights $\left(\frac{\mathrm{Var}[M_t]}{M}\right)$, it will not diverge and this bounded variance property is significant to preserve the concentration law for convergence guarantees.

### 5.5. Communication Bandwidth and Storage Analysis

The exchanging parameters in Algorithm 1 are online Q-table $\hat{Q}_i$ and updating number of visitation table $D_i^t$ (both matrix size $|S||A|$) for each connected agent pairs $i \in [M]$. The info exchange scheme uses pair-connection mode, which allows the exchange of only one pair of parameters. Therefore, the communication bandwidth is $\mathcal{O}(SA)$, which is the same level as other parameter sharing methods.

Parameters memory $\mathcal{P}$ can be created with fixed-size tensor initialization and is to store the above received parameters recursively: $2M$ parallel $|S||A|$ matrix, which can be compressed as a multi-dimensional tensor in calculation. The storage complexity $\mathcal{O}(MSA)$ is linear w.r.t. scaling up the system. This is also one of the core advances for Algorithm 1 that allows parallel computing that supports full-tensor computation in real large-scale engineering scenarios.

## 6. Theoretical Results Validations

The primary goal of the experimental validation is to demonstrate the effectiveness of our proposed AGC weights matrix for value aggregation under heterogeneous uncertainty in online decentralized MARL. Specifically, we conduct three sets of experiments: (i) to illustrate the learning rollback phenomenon formulated in Section 3 and compare it with the proposed AGC method; (ii) to evaluate the scalability of AGC with respect to the number of agents, corroborating the theoretical results in Theorem 5.6; (iii) to assess the robustness of AGC under different levels of random noise $M_t$. All experiments are performed in two simulated environments: an artificial MDP policy evaluation task and a cooperative navigation MPE task (Mordatch & Abbeel, 2018).

**Policy evaluation in symbolic MDP:** Consider a symbolic MDP $\mathcal{M} = \{\mathcal{S}, \mathcal{A}, P, r, \gamma\}$ with only one state $\mathcal{S} = \{s\}$ and one action $\mathcal{A} = \{a\}$. The transition and reward function are fixed. The optimal Q-value for this symbolic MDP is:

$$Q^*(s,a) \equiv \frac{r}{1-\gamma}.$$

Two agents with different local update frequencies (uncer-

tainty) want to evaluate the optimal Q-value for the fixed policy. Following the aggregation scheme in Figure 2, each agent shares their estimated Q-value and local update steps.

**Cooperative navigation MPE task:** each agent moves on a bounded 2D grid-world with action set $\mathcal{A}_i = \{\uparrow, \downarrow, \leftarrow, \rightarrow, -\}$. All agents aim to reach a specified target position at $s_{tar}$ within maximal horizon steps $H$. The system returns per-agent noisy rewards as the distance to the target changes:

$$r_{t,m}(s,a) = |s_{t+1,m}-s_{\mathrm{tar}}|-|s_{t,m}-s_{\mathrm{tar}}|+\epsilon,\ \mathbb{E}[r] \in [-1,1],$$

and we record the overall system reward as score for the episodic performance evaluation. In the networked dec-MARL setting, each agent exchanges their parameters ($Q$-tables and $D_i^t$-tables) with other agents through the network. The whole experiment is separated into training and validation phases. During training, all agents are initialized from fixed initial positions in the grid to learn an optimal Q-table that maximizes the system's episodic cumulated rewards, while at validation, the initial positions are swapped adversarially to verify whether they can still accomplish the task under the learned optimal Q-table in the test time. The detailed setup is given in Appendix E.1.

### 6.1. Learning Rollback vs AGC Performance

In the first set of experiments, we demonstrate the learning rollback phenomenon of traditional FedAVG in symbolic MDP value aggregation under heterogeneous uncertainty and compare it with AGC. This experiment isolates learning rollback caused solely by aggregating value estimates with unequal uncertainty in a controlled symbolic MDP setting.

With $r = 1, \gamma = 0.9$ ($Q^* = 10$), an agent performs $E_1 = 10$ local updates and aggregates with a less experienced agent ($E_2 \in \{0, 2, 5, 9\}$) after 20 rounds over 50 total rounds. Figure 3 shows the rollback under different uncertainty levels. Next, two agents periodically aggregate their Q-estimates using either convex averaging (FedAVG) or the proposed AGC weights. Figure 3 further compares the aggregation error and convergence speed, highlighting the effectiveness of AGC over traditional FedAVG. More performance comparisons on the MPE environment are discussed in Appendix E.2.

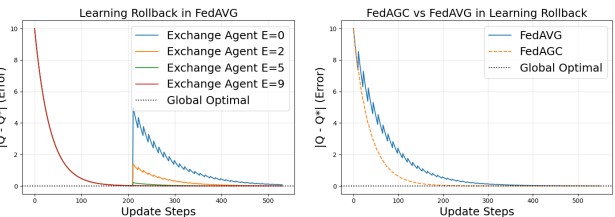

*Figure 3.* FedAVG Rollback (left) and AGC Performance (right).

## 6.2. Agent Scalability and Convergence Acceleration

In the second set of experiments, we evaluate the scalability of the proposed AGC method w.r.t. the number of agents. We let $M \in \{2, 4, 6, 8, 10, 12\}$ in the MPE cooperative navigation task with AGC-based value aggregation. Agents are initialized at the corners of a $10 \times 10$ grid-world to induce heterogeneous uncertainty in their value estimation, with the aim of reaching the central target located at $s_{tar} = (5, 5)$.

We fix $\epsilon = 0$, and train the system for $K = 1500$ episodes with horizon $H = 10$, global consensus is updated every $E = 5$ steps and discount factor $\gamma = 0.9$. The maximal system score under the optimal policy equals the the sum of the distances between each agent's initial position and the target, attained when all agents move directly towards the target without redundant actions. We normalize the $y$-axis by this maximum score to report the percentage of the collective task accomplished. Training results are presented in Figs. 4-5 show the relation between number of agents and the observed training episodes.

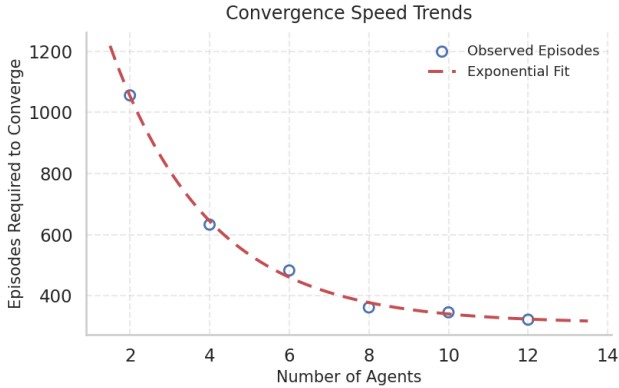

*Figure 4.* Convergence speed comparison with agent numbers.

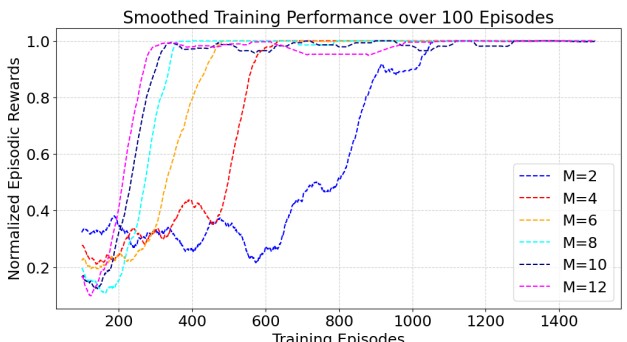

*Figure 5.* Training performance for scalability experiments.

## 6.3. Robustness to Stochastic Noise

In the third set of experiments, we use the same MPE environment settings as in Section 6.2, while fixing the num-

ber of agents to $M = 8$ and varying the noise level as $\text{std}(\epsilon) \in \{0.1, 0.2, 0.4, 0.6\}$. This experiment evaluates the robustness of the proposed AGC method against stochastic noise in the reward signal. The training curves are shown in Fig. 6, and the validation statistics are shown in Table 1.

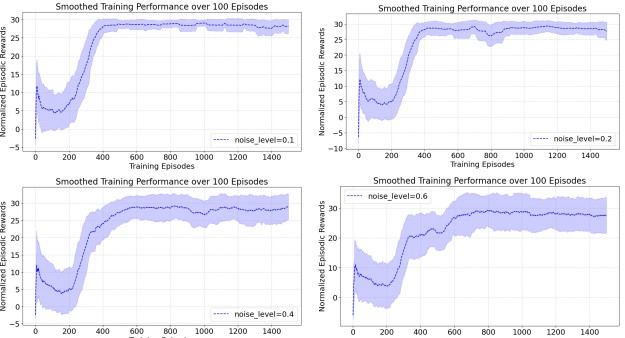

*Figure 6.* Training results with varying noise levels.

*Table 1.* Validation Statistics for Robustness over 8 Episodes.

| Noise level (std) | 0.1 | 0.2 | 0.4 | 0.6 |
|---|---|---|---|---|
| Score Mean | 28.19 | 28.88 | 28.60 | 24.41 |
| Score std | 0.75 | 1.76 | 3.74 | 6.81 |

## 6.4. Discussions

The first set of experiments illustrates the learning rollback phenomenon of the traditional FedAVG algorithm with convex weights, even in a symbolic single-state MDP. Figure 3 (left) shows that the rollback effect becomes less pronounced as the uncertainty gap between aggregated values decreases: specifically, the blue curve ($E = 0$, largest uncertainty difference from $E = 10$) exhibits the most severe rollback, whereas the red curve ($E = 9$, smallest uncertainty difference) shows a much milder effect when communication begins at round 20. Meanwhile, Fig. 3 (right) demonstrates that the proposed AGC method effectively eliminates learning rollback when agents ($E = 10$ and $E = 1$) communicate from the beginning of the training. These results empirically confirm that learning rollback in online decentralized MARL arises from aggregating value estimates with heterogeneous uncertainty, and that AGC successfully mitigates this issue through uncertainty-aware weighting.

The second set of experiments evaluates the scalability of the proposed AGC method. Figure 5 presents the smoothed training curves in the MPE environment for different numbers of agents, all of which achieve 100% collective task completion at test time. We record the first training episode at which convergence occurs in Fig. 4, revealing an approximately exponential relationship between convergence speed and number of agents. From $M = 2$ to $M = 12$, the time required to reach the optimal policy decreases rapidly. This

observation is consistent with Theorem 5.6, which states that AGC is equivalent to accumulating all shared local updates and evolving toward a lower-uncertainty estimate, leading to accelerated convergence.

The last set of experiments demonstrates the robustness of AGC under stochastic reward noise. For noise levels $std(\epsilon) \in \{0.1, 0.2, 0.4\}$, both the training curves and the validation statistics show that the average social performance remains close to the global optimum, while the variance of the scores increases with the noise level. As a stress test, we further consider a high-noise regime where the noise level reaches 60% of the reward signal; even in this case, agents manage to converge and accomplish the MPE task. Moreover, the performance variance is nonlinear but can be amplified w.r.t. the noise level. This behavior aligns with the variance bound analysis in Theorem 5.6, validating the theoretical robustness of the proposed AGC method.

## 7. Related Work

Classical MARL approaches often rely on a centralized training that observes the global state and outputs joint actions for all agents, such as MADDPG (Lowe et al., 2017), MAPPO (Yu et al., 2022), and COMA (Foerster et al., 2018). While effective in small systems, centralized methods suffer from poor scalability due to the exponential growth of the joint state–action space (Ma et al., 2024; Chu et al., 2019) and are unsuitable for partially observable settings (Liu et al., 2022; Omidshafiei et al., 2017). To address these challenges, **decentralized MARL (dec-MARL)** has gained popularity (de Witt et al., 2020; Jiang & Lu, 2022; Mao et al., 2022; Hairi et al., 2024; Gabler & Wollherr, 2024). In dec-MARL, agents independently learn from local interactions without global coordination, improving scalability and reducing sample complexity in the number of agents from exponential to linear (Su et al., 2022; Zhang et al., 2018b).

However, fully independent dec-MARL often fails to capture essential inter-agent dependencies, leading to unstable or suboptimal learning (Jiang & Lu, 2022; Zhang et al., 2021a; 2018b). To enhance coordination, **networked dec-MARL** frameworks (Zhang et al., 2018b; Chu et al., 2020; Zhang et al., 2021b) introduce structured communication networks that enable information exchange among agents. Depending on the exchanged content, agents may share past experience data or learned parameters (Albrecht et al., 2024). Sharing experience data (Chu et al., 2025; Lidard et al., 2022) improves sample efficiency by increasing updates per timestep, but it raises privacy concerns, increases memory usage, and requires reliable network connectivity.

To mitigate these issues, recent work explores **parameter sharing** as a communication-efficient alternative (Zhang

et al., 2018b; Gupta et al., 2017; Zhang et al., 2018a; Sunehag et al., 2018; Kim & Sung, 2023; Wen et al., 2021; Li et al., 2023; 2020a). This idea naturally connects to **federated learning** (FL), where distributed agents collaboratively optimize a shared model while keeping local data private, such as FedAvg (Konečnỳ et al., 2016; McMahan et al., 2017) and FedProx (Li et al., 2020b). In networked multi-agent reinforcement learning (MARL), this perspective has led to federated variants of value-based methods, such as Fed-Q Learning (Zhuo et al., 2019), which aggregates Q-values across agents to accelerate decentralized learning without raw data exchange. These approaches are designed to achieve global parameter consensus in multi-agent systems and often come with convergence and sample complexity guarantees in offline setting with uniformly ergodicity.

More recent work has extended FL-style analysis to settings with agent heterogeneity. (Reddi et al., 2020) study the impact of heterogeneity on communication efficiency, while (Wang et al., 2020) and (Jin et al., 2022) analyze FedAvg and FedProx under objective and environment heterogeneity. Subsequent studies establish convergence guarantees for heterogeneous Fed-Q learning for discrete state-action space and continuous linear function approximation settings (Wang et al., 2023; 2025). However, most of these results still rely on offline learning assumptions or uniform ergodicity. In contrast, practical online learning in networked MARL violates these assumptions, where aggregation strategies such as FedAvg may suffer from degraded convergence behavior and even instability under heterogeneity (Qi et al., 2021; Li et al., 2019).

## 8. Conclusion and Future Work

In this paper, we formally characterize the learning rollback phenomenon for value aggregation in online dec-MARL. Our proposed AGC method overcomes this issue by dynamically adjusting weights based on uncertainty. For future research, one promising avenue is to explore adaptive weighting in settings involving non-linear operators, enabling broader theoretical generalization of stochastic dynamics. Another direction is to investigate the algorithm's performance over more complex network structures, to enhance scalability and resilience. Additionally, integrating adaptive consensus into hierarchical or heterogeneous MARL systems could further improve coordination efficiency and learning stability in large-scale applications.

## Impact Statement

This paper aims to advance the field of MARL. Due to the impact of this area in modern multi-agent systems, including LLMs and agentic AI, there can be many potential societal consequences, but it is difficult to pinpoint specific ones.

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

## A. Online Networked Dec-MARL Framework

A detailed online networked dec-MARL framework with the Decentralized Training Decentralized Execution (DTDE) workflow of Figure 2 is shown in Algorithm 2 below:

---
**Algorithm 2** Online Networked Dec-MARL Framework

---
**Require:** Environment $E$, network $\mathcal{G}([M], \mathcal{E})$
**Ensure:** Learned action beliefs $\{A_i\}_{i=1}^{M}$
1: Initialize local action belief
 ▷ *Either Q-estimation or policy parameters initialization*
2: **for** each episode $k = 1$ to $K$ **do**
3:   Reset environment, get initial state $s_{0,m}$
4:   **for** each timestep $h \in [H]$ **do**
5:     **for** each agent $m \in [M]$ **do**
 ▷ *Dec-execution, collect online data and process update info*
6:       Select action $a_{h,m}^{k}$ from its action belief
7:       Execute $a_{h,m}^{k}$, observe reward $r_{h,m}^{k}$, next state $s_{h+1,m}^{k}$
8:       Pass and receive update info from neighbors $\mathcal{N}_m$
9:     **end for**
10:     Update each agent's action belief via collected info
11:   **end for**
12: **end forreturn** Agents set $\{\mathcal{A}_i\}_{i=1}^{M}$ with learned action belief

---

## B. AGC Weights in an Illustrative 2-Agent Single-state MDP Example

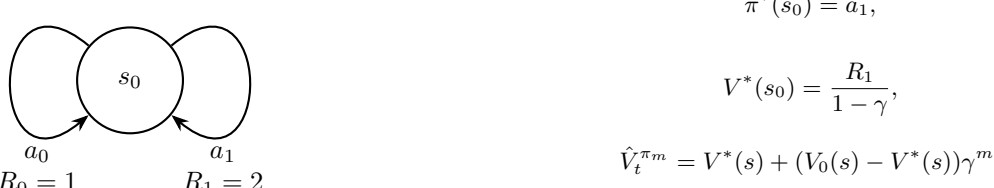

$$\pi^*(s_0) = a_1,$$

$$V^*(s_0) = \frac{R_1}{1-\gamma},$$

$$\hat{V}_t^{\pi_m} = V^*(s) + (V_0(s) - V^*(s))\gamma^m.$$

*Figure 7.* A toy example of a single-state MDP.

Consider a single-state MDP $\mathcal{M} = (\mathcal{S}, \mathcal{A}, \{\mathbb{P}_h\}_{h=1}^{H}, \mathcal{R}_h, \gamma)$, where $s_0 \in \mathcal{S}$ is the only state, the action set is $\mathcal{A} = \{0, 1\}$, and $\gamma$ is the discount factor. This deterministic MDP provides reward $R_0 = 1$ for action $a_0$ and $R_1 = 2$ for action $a_1$, as shown in Figure 7. Action $a_1$ is advantageous, so the optimal policy is $\pi^* = a_1$, with the optimal state value:

$$V^*(s_0) = Q^*(s_0, a_1) = \lim_{H \to \infty} \sum_{h=1}^{H} \gamma^{h-1} R_1 = \frac{R_1}{1-\gamma}.$$

Now consider two strongly homogeneous agents, $M$ and $N$, learning the optimal state value in parallel, with $m$ and $n$ local iterations, respectively. They share their estimated values $\hat{V}_t^{\pi_m}$ and $\hat{V}_t^{\pi_n}$ every $E$ steps as in Algorithm 2. For this fully connected two-agent setting, the global consensus reduces to a pair-wise parameter update. When $m \neq n$, the consensus weights are:

$$\begin{cases} w_m = \dfrac{\gamma^n(1-\gamma^m)}{\gamma^n - \gamma^m}, \\ w_n = \dfrac{-\gamma^m(1-\gamma^n)}{\gamma^n - \gamma^m}. \end{cases} \tag{16}$$

The updated value after the global consensus step is

$$\hat{V}_{t'}^{\pi_m} = \hat{V}_{t'}^{\pi_n} = w_m \hat{V}_t^{\pi_m} + w_n \hat{V}_t^{\pi_n}. \tag{17}$$

**B.1. Convergence Analysis in the 2-Agent Setting**

Assuming the discount factor $\gamma$ is known, Algorithm 1 converges for the 2-agent policy evaluation. For any state $s \in \mathcal{S}$ with initialization $V_0$, each agent's estimate is

$$\hat{V}_t^{\pi_i} = V^*(s) + (V_0(s) - V^*(s))\gamma^i, \quad i \in \{m, n\},$$

where $V^*(s) = V^{\pi_m^*} = V^{\pi_n^*}$. Applying equation (17), after one global consensus step:

$$\hat{V}_{t'}^{\pi_i}(s) = V^*(s) + (V_0(s) - V^*(s))\gamma^{m+n}, \quad \forall i \in \{m, n\}. \tag{18}$$

This holds because $w_m + w_n = 1$, ensuring convergence to $V^*(s)$, while extrapolation weights ($\exists i \in \{m, n\}, |w_i| > 1$) extend the learning iteration from $\gamma^i$ to $\gamma^{m+n}$.

In the special case $m = n$, the two agents have equal local experience, and the consensus reduces to simple federated averaging:

$$\hat{V}_{t'}^{\pi_i}(s) = \frac{1}{2}(\hat{V}_t^{\pi_m}(s) + \hat{V}_t^{\pi_n}(s)) = V^*(s) + (V_0(s) - V^*(s))\gamma^i, i = m = n.$$

The analysis demonstrates that the designed weights in equations (16) and (18) ensure Algorithm 1 converges to the true optimal policy evaluation, and convergence is accelerated by incorporating updates from the other agent.

## C. Relation between $\mathrm{Pow}(D_i^t)$ and Uncertainty in Equation 13

In the context of online policy evaluation, the Bellman contraction property of TD(0)-style Q-value updates (see Lemma 4.1) implies that the estimation error between the learned Q-values and the target Q-values decays geometrically with the number of state-action visitations. Specifically, the corresponding upper bound can be characterized by powers of the discount factor $\gamma$, i.e., terms of the form $\gamma^k$, where $k$ denotes the number of effective updates or visitations.

We therefore use these $\gamma$-power terms to represent the uncertainty (or residual bias) in online policy evaluation. Since TD-style temporal-difference updates constitute the fundamental learning dynamics underlying many modern RL algorithms, this characterization naturally extends to a broad class of RL methods for evaluating learning uncertainty and convergence behavior (Woo et al., 2025). Such quantities are rooted in RL learning theory and are widely used in the analysis of convergence rates and sample complexity, providing a formal statistical measure of learning confidence.

Based on this characterization, $\mathrm{Pow}(D_i^t)$ serves as a matrix representation of the learning uncertainty over the entire state-action space, defined as

$$\mathrm{Pow}(D)(s, a) = \gamma^{D(s,a)},$$

where $D(s, a)$ denotes the number of local state-action updates between two aggregation steps at state-action pair $(s, a)$. Since $\gamma^{D(s,a)}$ quantifies the remaining uncertainty after local learning updates, $\mathrm{Pow}(D)$ captures the confidence gain accumulated between consecutive aggregations. Our AGC method constructs aggregation weights directly from this confidence gain, enabling the aggregation process to consistently favor more reliable updates and thereby guarantee convergence through continuous reduction of estimation bias and uncertainty.

## D. Proofs

**D.1. Proof of Lemma 2.3**

*Proof.* Suppose for any agent $i \in [M]$, their estimated Q-values uniform converges to the optima $Q^*$, i.e. $\forall \epsilon > 0$, there exists an inputs $(s_i, a_i)$-irrelevant function $T(\epsilon)$, such that

$$|\hat{Q}_{t,i} - Q^*| < \epsilon, \forall i \in [M], (s, a) \in \mathcal{S} \times \mathcal{A}, t > T(\epsilon).$$

Then, at any $(s, a)$, for any $t > T(\epsilon)$

$$|\hat{Q}_{t',m} - Q^*| = |\sum_{i \in [M]} w_{i,m}\hat{Q}_{t,i} - Q^*| \overset{\textcircled{1}}{=} |\sum_{i \in [M]} w_{i,m}(\hat{Q}_{t,i} - Q^*)| < \sum_{i \in [M]} w_{i,m}|\hat{Q}_{t,i} - Q^*| < \epsilon$$

where step $\textcircled{1}$ considers convex weights property. The inequality holds for all $t' > T(\epsilon)$, so the uniform convergence $\hat{Q}_{t',m} \rightrightarrows Q^*$ with $t' = t$ is proved. $\square$

## D.2. Proof of Uniform Convergence in Equation 8 & Proposition 3.1

Following equation (5):

$$|(\hat{Q}_{t,i} - Q^*)(s,a)| < \epsilon_i, \forall t > T(\epsilon_i, s, a),\ i \in [M].$$

Condition $||(\hat{Q}_{t',m} - Q^*)||_\infty < ||\hat{Q}_{t,i} - Q^*||_\infty, \forall i, m \in [M]$ implies that for any fixed $(s,a) \in \mathcal{S} \times \mathcal{A}$,

$$|(\hat{Q}_{t',m} - Q^*)(s,a)| < |(\hat{Q}_{t,i} - Q^*)(s,a)| < \epsilon_i, \tag{19}$$

all holds for all agents $i \in [M]$. We can get:

$$|(\hat{Q}_{t',m} - Q^*)(s,a)| < \min_i \{\epsilon_i\} = T^{-1}(\max_i \{t_i(s,a)\}) < T^{-1}(t) = \epsilon, \tag{20}$$

Therefore, for any arbitrarily small $\epsilon$, we can always find such state-action independent timestep $T(\epsilon)$, such that for any fixed $i \in [M], (s,a) \in \mathcal{S} \times \mathcal{A}$, the following uniform convergence condition holds:

$$|(\hat{Q}_{t',m} - Q^*)(s,a)| < \epsilon, \forall t > T(\epsilon), \tag{21}$$

which proves uniform convergence of the aggregated outcome $\hat{Q}_{t',m}$.

Note that in the proof, we did not use the properties of convex weights, and in fact, this requirement in Equation 20 contradicts with the result of convex properties in Equation 7. Therefore, no solution can be found given the 2 conflicted constraints, and Proposition 3.1 is proved.

## D.3. Proof of Lemma 5.1 & Lemma 5.2

*Proof.* We consider the AGC weights $W_m^t(s,a), W_{i,m}^t(s,a)$ at any fixed $(s,a)$ and $m \in [M]$, where $D_m^t(s,a) \neq D_i^t(s,a)$.

To prove Lemma 5.1, we show $\sum_{i \in \mathcal{N}_m} W_{i,m}^t(s,a) + W_m^t(s,a) =$

$$\sum_{i \in \mathcal{N}_m} \frac{\text{Pow}(D_i^t)(s,a) - \text{Pow}(D_m^t + \sum_j D_j^t)(s,a)}{A_m(s,a)(\text{Pow}(D_i^t)(s,a) - \text{Pow}(D_m^t)(s,a))} - \sum_{i \in \mathcal{N}_m} \frac{\text{Pow}(D_m^t)(s,a) - \text{Pow}(D_m^t + \sum_j D_j^t)(s,a)}{A_m(s,a)(\text{Pow}(D_i^t)(s,a) - \text{Pow}(D_m^t)(s,a))}$$

$$= \frac{1}{A_m(s,a)} \sum_{i \in \mathcal{N}_m} \frac{\text{Pow}(D_i^t)(s,a) - \text{Pow}(D_m^t + \sum_j D_j^t)(s,a) - \text{Pow}(D_m^t)(s,a) + \text{Pow}(D_m^t + \sum_j D_j^t)(s,a)}{\text{Pow}(D_i^t)(s,a) - \text{Pow}(D_m^t)(s,a)}$$

$$= \frac{A_m(s,a)}{A_m(s,a)} = 1.$$

We can further show this normalization lemma still holds for the place where $D_m^t(s,a) = D_i^t(s,a)$, since the weights there applies equal weights distribution that sums up to 1.

To prove Lemma 5.2, according to the definition of $W_{i,m}^t$, we have for each AGC weights:

$$W_{i,m}^t(s,a) = -\frac{\text{Pow}(D_m^t)(s,a) - \text{Pow}(D_m^t + \sum_j D_j^t)(s,a)}{A_m(s,a)(\text{Pow}(D_i^t)(s,a) - \text{Pow}(D_m^t)(s,a))} = \frac{1}{A_m(s,a)} \frac{\text{Pow}(D_m^t)(s,a)(1 - \text{Pow}(\sum_j D_j^t)(s,a))}{\text{Pow}(D_m^t)(s,a)(1 - \text{Pow}(D_i^t - D_m^t)(s,a))}$$

$$\overset{\textcircled{1}}{\leq} \frac{1}{A_m(s,a)} \frac{1 - \gamma^{ME}}{1 - \gamma} \leq \frac{1 - \gamma^{ME}}{1 - \gamma} = Z.$$

where the inequality $\textcircled{1}$ holds because the numerator is always positive and achieves maximum when $\sum_j D_j^t(s,a) \leq A_m(s,a)E \leq ME$ achieves maximum, while the denominator achieves positive minimum to get the upper bound when $D_i^t \geq D_m^t + 1$. Similarly, we can get the symmetric lower bound for $W_{i,m}^t(s,a) \geq \frac{\gamma^{ME}-1}{1-\gamma}$. Recall the normalized constraint of the sum among $W_m^t$ and all $W_{i,m}^t$ in Lemma 5.1, we can quickly get the range of $W_m^t(s,a), \forall (s,a)$. Here, we denote $\max |W_{i,m}^t(s,a)| = \frac{1-\gamma^{ME}}{1-\gamma} = Z$ for later use. We can also easily verify that when $D_m^t(s,a) = D_i^t(s,a)$, the equal weights are $1/A_m(s,a)$, which still lie in in this Lemma bound. $\square$

**D.4. Proof of Lemma 5.3**

*Proof.* We first prove this lemma by showing the equivalence of the designed AGC weights matrix $W_{i,m}^t$ for terminal agent $i \in \mathcal{N}_m$. At any fixed $(s,a)$ and $m \in [M]$, where $D_m^t(s,a) \neq D_i^t(s,a)$, recall the definition of terminal agents $i$'s weight $W_{i,m}^t(s,a)$ in equation (13):

$$A_m(s,a)W_{i,m}^t(s,a) = -\frac{\text{Pow}(D_m^t)(s,a) - \text{Pow}(D_m^t + \sum_j D_j^t)(s,a)}{\text{Pow}(D_i^t)(s,a) - \text{Pow}(D_m^t)(s,a)}.$$

Its numerator can be seen as an agent-invariant constant for all terminal agents $i$ in the recursive computation, since both the target agent's local update number change $D_m^t$ and the system's total $D_m^t + \sum_{j \in \mathcal{N}_m} D_j^t$ are invariant to each $i$. The system's total updates number $D_m^t + \sum_{j \in \mathcal{N}_m} D_j^t$ is calculated recursively by the matrix accumulator $D_m^{sum}$ in the decentralized setting, where each terminal only contributes its local update number once when connected with the target to exchange the parameters in parallel. The variant denominator part of the terminal weights matrix $\text{Pow}(D_m^t)(s,a)/(\text{Pow}(D_i^t)(s,a) - \text{Pow}(D_m^t)(s,a))$ is stored in a 2D-list $\mathcal{P}$ as the first column element with the corresponding shared parameter $\hat{Q}_i$ being the second. At the end of exchanging parameters with all connected terminals, the recursive updated results are:

$$\text{Accumulators } D_m^{sum}(s,a) = \sum_j D_j^t(s,a), \ A_m(s,a) = \sum_{i \in \mathcal{N}_m} \mathbb{I}[D_i^t(s,a) \neq D_m^t(s,a)].$$

$$\text{Inner product of the list } \mathcal{P}\text{'s elements:} \langle \mathcal{P}_0, \mathcal{P}_1 \rangle(s,a) = \sum_{i \in \mathcal{N}_m} \frac{\text{Pow}(D_m^t)(s,a) \cdot \hat{Q}_i(s,a)}{\text{Pow}(D_i^t)(s,a) - \text{Pow}(D_m^t)(s,a)}.$$

At the parameter exchanging phase, target agent $m$ also transfers its error range $\text{Pow}(D_m^t)$ to all terminal agents, so the adaptive weights for each terminal agent $i$ are:

$$W_{i,m}^t(s,a) = -\frac{1}{A_m(s,a)} \frac{\text{Pow}(D_m^t)(s,a) - \text{Pow}(D_m^t + \sum_j D_j^t)(s,a)}{\text{Pow}(D_i^t)(s,a) - \text{Pow}(D_m^t)(s,a)} = \frac{1}{A_m(s,a)} \frac{\text{Pow}(D_m^{sum})(s,a) - 1}{\text{Pow}(D_i^{dif})(s,a) - 1},$$

where $D_i^{dif}(s,a) = D_i^t(s,a) - D_m^t(s,a)$ is computed with the transmitted $D_m^t(s,a)$ from target agent $m$ and its $D_i^t(s,a)$.

Considering the fact from Lemma 5.1, we can quickly get the target agent's weight $W_m^t$ from the normalization constraint:

$$W_m^t(s,a) = 1 - \sum_{i \in \mathcal{N}_m} W_{i,m}^t(s,a) = 1 - \frac{\sum_i \mathcal{P}_0(s,a)(\text{Pow}(D_m^{sum})(s,a) - 1)}{A_m(s,a)}.$$

For cases $D_m^t(s,a) = D_i^t(s,a)$, we can still apply the same parallelized matrix learning algorithm by resolving its numerical issues in the numerator, but overwrites all such values with the equal weights distribution as in Equation (14).

Considering communication delay in practice, where the system-level data is not available, this computation avoids a direct centralized parameters collection, realizing a fully decentralized global consensus computation for any $(s,a) \in \mathcal{S} \times \mathcal{A}$:

$$\hat{Q}_m(s,a) \leftarrow W_m^t(s,a)\hat{Q}_m(s,a) + \sum_{i \in \mathcal{N}_m} W_{i,m}^t(s,a)\hat{Q}_i(s,a) = W_m^t(s,a)\hat{Q}_m(s,a) + \langle [W_{i,m}^t](s,a), \mathcal{P}_1(s,a) \rangle,$$

which completes the proof for Lemma 5.3. $\qquad \square$

**D.5. Proof of Theorem 5.4**

*Proof.* For any fixed $(s,a) \in \mathcal{S} \times \mathcal{A}$ with $D_i^t(s,a) \neq D_m^t(s,a)$, without loss of generality, suppose all agents update their local update number at $(s,a)$ to $N_m^t(s,a)$ after last global consensus making (will be proved in Theorem 5.6) and $N_m^0(s,a) = 0, \forall (s,a)$ on the initialization. The updates for learned parameters $\hat{Q}$ are all from local collected data as in subsection 4.1 since last global updates, then for all such $(s,a), i \in \mathcal{N}_m$,

$$\begin{cases} \mathbb{E}[\hat{Q}_m(s,a)] = \mathbb{E}[\mu_m] + \text{Pow}(N_m^t + D_m^t)(s,a)\mathbb{E}[\alpha_m], \\ \mathbb{E}[\hat{Q}_i(s,a)] = \mathbb{E}[\mu_i] + \text{Pow}(N_m^t + D_i^t)(s,a)\mathbb{E}[\alpha_i]. \end{cases}$$

For the homogeneous agents in Assumption 2.1, we can get $\mathbb{E}[\mu_m] = \mathbb{E}[\mu_i], \forall i \in [M]$ at the given $(s, a)$, where $D_i^t(s, a) \neq D_m^t(s, a)$. Solving the above linear equations we get:

$$
\begin{cases}
\mathbb{E}[\mu_m] = \dfrac{\text{Pow}(D_i^t)(s, a)\mathbb{E}[\hat{Q}_m(s, a)] - \text{Pow}(D_m^t)(s, a)\mathbb{E}[\hat{Q}_i(s, a)]}{\text{Pow}(D_i^t)(s, a) - \text{Pow}(D_m^t)(s, a)} = \mathbb{E}[\hat{\mu}_{m,i}], \\[3mm]
\mathbb{E}[\alpha_m] = \dfrac{\mathbb{E}[\hat{Q}_m(s, a)] - \mathbb{E}[\hat{Q}_i(s, a)]}{\text{Pow}(N_m^t)(s, a)(\text{Pow}(D_m^t)(s, a) - \text{Pow}(D_i^t)(s, a))} = \mathbb{E}[\hat{\alpha}_{m,i}],
\end{cases}
$$

which completes the proof. $\qquad\square$

### D.6. Proof of Theorem 5.6

*Proof.* Recall the global update formula in equation (13) and (15), and consider the following accelerated Q-estimation quantity according to Corollary 5.5 and Theorem 5.4 for any given $(s, a)$ with $D_m^t(s, a) \neq D_i^t(s, a)$:

$$
\mathbb{E}[\mu_m] + \mathbb{E}[\alpha_m]\,\text{Pow}(D_m^t + \sum_{i \in \mathcal{N}_m} D_i^t)(s, a) = \mathbb{E}[\bar{\mu}_m] + \mathbb{E}[\bar{\alpha}_m]\,\text{Pow}(D_m^t + \sum_{i \in \mathcal{N}_m} D_i^t)(s, a)
$$

$$
= \frac{1}{A_m(s, a)}\sum_{i \in \mathcal{N}_m}\left[\frac{\text{Pow}(D_i^t)(s, a)\mathbb{E}[\hat{Q}_m(s, a)] - \text{Pow}(D_m^t)(s, a)\mathbb{E}[\hat{Q}_i(s, a)]}{\text{Pow}(D_i^t)(s, a) - \text{Pow}(D_m^t)(s, a)}\right.
$$

$$
\left. + \frac{\mathbb{E}[\hat{Q}_m(s, a)] - \mathbb{E}[\hat{Q}_i(s, a)]}{\text{Pow}(N_m^t)(s, a)(\text{Pow}(D_m^t)(s, a) - \text{Pow}(D_i^t)(s, a))}\,\text{Pow}(N_m^t + D_m^t + \sum_{i \in N_m} D_i^t)(s, a)\right]
$$

$$
= \left(\frac{1}{A_m(s, a)}\sum_{i \in \mathcal{N}_m}\frac{\text{Pow}(D_i^t)(s, a) - \text{Pow}(D_m^t + \sum_j D_j^t)(s, a)}{\text{Pow}(D_i^t)(s, a) - \text{Pow}(D_m^t)(s, a)}\right)\mathbb{E}[\hat{Q}_m(s, a)]
$$

$$
+ \sum_{i \in \mathcal{N}_m}\frac{1}{A_m(s, a)}\frac{\text{Pow}(D_m^t + \sum_j D_j^t)(s, a) - \text{Pow}(D_m^t)(s, a)}{\text{Pow}(D_i^t)(s, a) - \text{Pow}(D_m^t)(s, a)}\mathbb{E}[\hat{Q}_i(s, a)]
$$

$$
= \mathbb{E}[W_m^t(s, a)\hat{Q}_m(s, a) + \sum_{i \in \mathcal{N}_m} W_{i,m}^t(s, a)\hat{Q}_i(s, a)] = \mathbb{E}[\hat{Q}_m]_{new},
$$

which completes the proof for expectation part. Next, let's consider the variance of global updated $\hat{Q}_m(s, a)_{new}$:

$$
\text{Var}[\hat{Q}_m(s, a)]_{new} = \mathbb{E}\left[(W_m^t(s, a)\hat{Q}_m(s, a) + \sum_{i \in \mathcal{N}_m} W_{i,m}^t(s, a)\hat{Q}_i(s, a) - E[\hat{Q}_m(s, a)_{new}])^2\right]
$$

$$
= \mathbb{E}\left[(W_m^t(s, a) * M_{t,m} + \sum_{i \in \mathcal{N}_m} W_{i,m}^t(s, a) * M_{t,i})^2\right] = \text{Var}[M_t]\left(\frac{W_m^t(s, a)^2}{|\mathcal{N}_m| - A_m(s, a) + 1} + \sum_i W_{i,m}^t(s, a)^2\right),
$$

For the variance lower bound, using Cauchy–Schwarz Inequality:

$$
\frac{W_m^t(s, a)^2}{|\mathcal{N}_m| - A_m(s, a) + 1} + \sum_i W_{i,m}^t(s, a)^2 \geq \frac{(\sum_{i \in \mathcal{N}_m} W_{i,m}^t(s, a) + W_m^t(s, a))^2}{|\mathcal{N}_m| - A_m(s, a) + 1 + \sum_{i \in \mathcal{N}_m} \mathbb{I}[D_m^t(s, a) \neq D_i^t(s, a)]} \geq \frac{1}{M}.
$$

For the variance upper bound, recall range for the designed weights in Lemma 5.2,

$$
\frac{W_m^t(s, a)^2}{|\mathcal{N}_m| - A_m(s, a) + 1} + \sum_{i \in \mathcal{N}_m} W_{i,m}^t(s, a)^2 \leq \frac{(1 + Z)^2}{|\mathcal{N}_m| - A_m(s, a) + 1} + \frac{Z^2}{A_m(s, a)} \overset{②}{\leq} (Z + 1)^2 + \frac{Z^2}{M - 1},
$$

where $Z = \frac{1 - \gamma^{ME}}{1 - \gamma}$, inequality ② holds because the left-hand side is a convex function w.r.t. $A_m(s, a)$ and its maximum achieved at its left boundary when $A_m(s, a) = |\mathcal{N}_m|$. $\qquad\square$

# E. Experiment Details for Section 6

## E.1. Simulated MPE for Cooperative Navigation Task Setups

We consider a $10 \times 10$ 2D grid world. The state space $\mathcal{S}$ is represented as an $M \times 2$ array, where each row corresponds to an agent's position $(x, y)$ within the grid (with each coordinate in $[0, 9]$). Let the action set for each agent be: $\mathcal{A}_i = \{\uparrow, \downarrow, \leftarrow, \rightarrow, -\}$, which corresponds to movements in the four cardinal directions (by one unit) or remaining still. Let the maximal horizon steps per episode $H = 10$, and all agents aim to reach a specified target position in the middle $s_{\text{tar}} = (5, 5)$. Agents initial positions are fixed. The environment returns the per-agent noisy reward as the change in distance to the target:

$$r_{t,m}(s, a) = |s_{t+1,m} - s_{\text{tar}}| - |s_{t,m} - s_{\text{tar}}| + \epsilon, \quad r \in [-1, 1],$$

and we record the overall system reward as score for the episode for performance evaluation. Here, we introduce a gaussian noise $\epsilon$ with controllable variance and zero mean. In our fully decentralized networked MARL setting, an adjacency matrix encodes the connection among agents. Each agent exchanges their parameters ($Q$-tables and $\Delta N$-tables) with the opponent agent through the network. Figure 8 shows an visualized setup of 2 agents in the MPE environment.

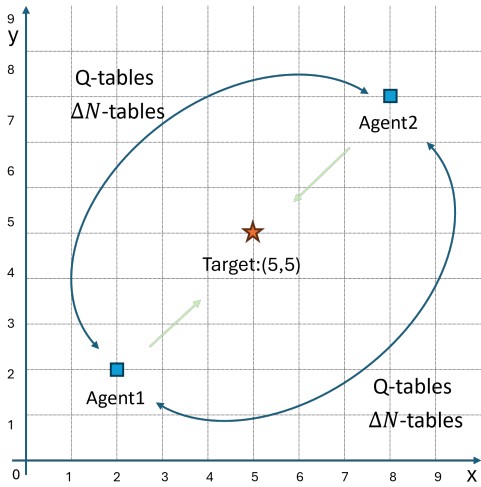

*Figure 8.* Simulated MPE for Cooperative Navigation Task

## E.2. Extended Experiment for Section 6.1

To further show the effectiveness of our proposed AGC method in online value aggregation with uncertainty, we compare the learning performance of 3 algorithms for online networked dec-MARL framework: (1) value aggregation with simple averaging (FedAVG), with (2) our proposed Adaptive Global Consensus weights (FedAGC), and (3) direct experience data sharing (DataSharing). For parameter sharing methods, FedAVG applies simple convex weights ($\frac{1}{2}$) updating the global consensus parameters, while FedAGC applies our proposed AGC weights. Direct data sharing stacks all received data from the opponent agent and update its Q-table with the stacked experience each timestep.

*Table 2.* Validations (Statistical Analysis over 8 Episodes).

| Algorithms | FedAGC | FedAVG | DataSharing |
|---|---|---|---|
| Mean Score | 8.485 | 2.243 | 3.909 |

Figure 9 shows the training performance for the three algorithms under consideration. Our proposed FedAGC algorithm (red curve) outperforms in all three algorithms with a faster convergence to the global optimal score ($\sim$8.5) within around 1500 training episodes. Direct data sharing method (blue curve) approaches the global optimal score, but it will take longer episodes compared to FedAGC. Meanwhile, the traditional FedAVG (yellow curve) suffers from severe learning rollback issue and fails to converge at the optimal policy, which is our motivation to propose the FedAGC algorithm.

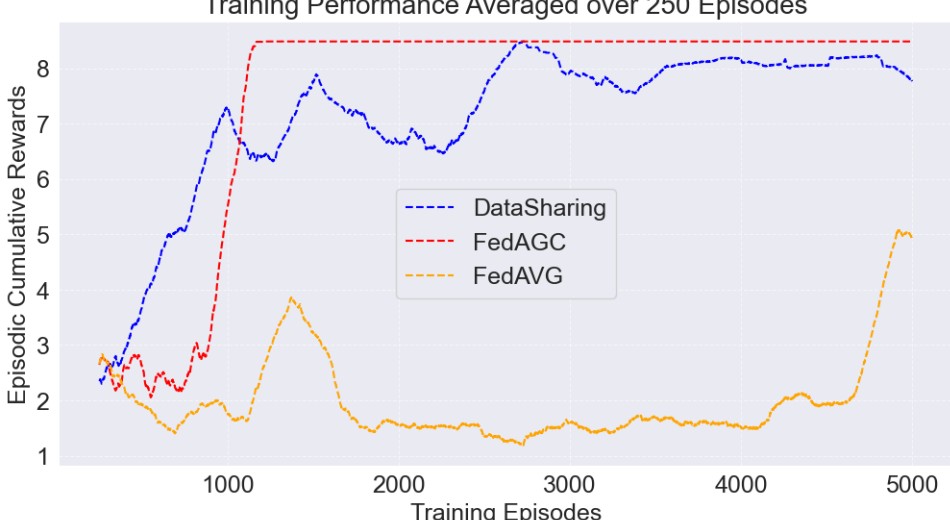

*Figure 9.* Training performance of our proposed FedAGC with comparison with traditional FedAVG and direct DataSharing method.

Table 2 presents the validation results after 5000 training episodes. In the test time, our FedAGC algorithm still achieves the global optimal mean of score over 8 validation episodes ($\sim 8.485$), which shows the optimal policy gained in training phase does not overfit. However, the validation results show that both the direct data sharing and FedAVG algorithms cannot accomplish all the tasks in the 8 validation episodes.

Additionally, we are considering 2 more Fed-style benchmarks, FedADP_lin (McMahan et al., 2017) and FedADP_exp (Wu & Wang, 2021), to show the universal existence of learning rollback issue in convex aggregations and compare the advances of our proposed FedAGC method. Formally, the normalized convex weights for 'FedADP_lin' and 'FedADP_exp' are proportional to the number of visits and exponential of number of visits respectively:

$$\text{FedADP\_lin: } Q(s,a) = \sum_{k=1}^{K} \frac{N_k(s,a)}{\sum_{j=1}^{K} N_j(s,a)} Q_k(s,a)$$

$$\text{FedADP\_exp: } Q(s,a) = \sum_{k=1}^{K} \frac{\exp(N_k(s,a))}{\sum_{j=1}^{K} \exp(N_j(s,a))} Q_k(s,a).$$

These two methods are popular adaptive Fed-style learning weights w.r.t. number of visitations. Figures 10, 11 shows additional experiment results on other Fed-style benchmarks.

The first set of experiment (Figure 10) shows that even adaptive Fed-learning will still face learning rollback issue in the online symbolic MDP setting, as can be seen from small rollbacks in the Zoomed-in view of the learning process. Compared to simple average weights, both FedADP_lin (green curve) and FedADP_exp (red curve) can accelerate the learning process, and FedADP_exp's performance tends to approach our proposed FedAGC (orange curve), but the learning rollback can not be avoided. The lower subfigure shows the zoomed-in view of the advances of FedAGC over FedADP_exp (the error of FedADP_exp is always larger than FedAGC, their difference is always positive), though their performances look similar.

The second set of experiment (Figure 11) is a complementary version of the early experiment in this section (Figure 9), where we are training 2 agents to complete a cooperative navigation task. For small local update frequency (we choose $E = 5$), the weights for FedADP_exp (orange line) and FedADP_lin (blue dotted) are similar, and we can observe their training performance (orange and blue curve) are similar in the training plot. Although both FedADP_exp and FedADP_lin manage to converge to the global optima, their convergence rate is slowed down due to the learning rollback issue compared to our proposed FedAGC (pink curve) method.

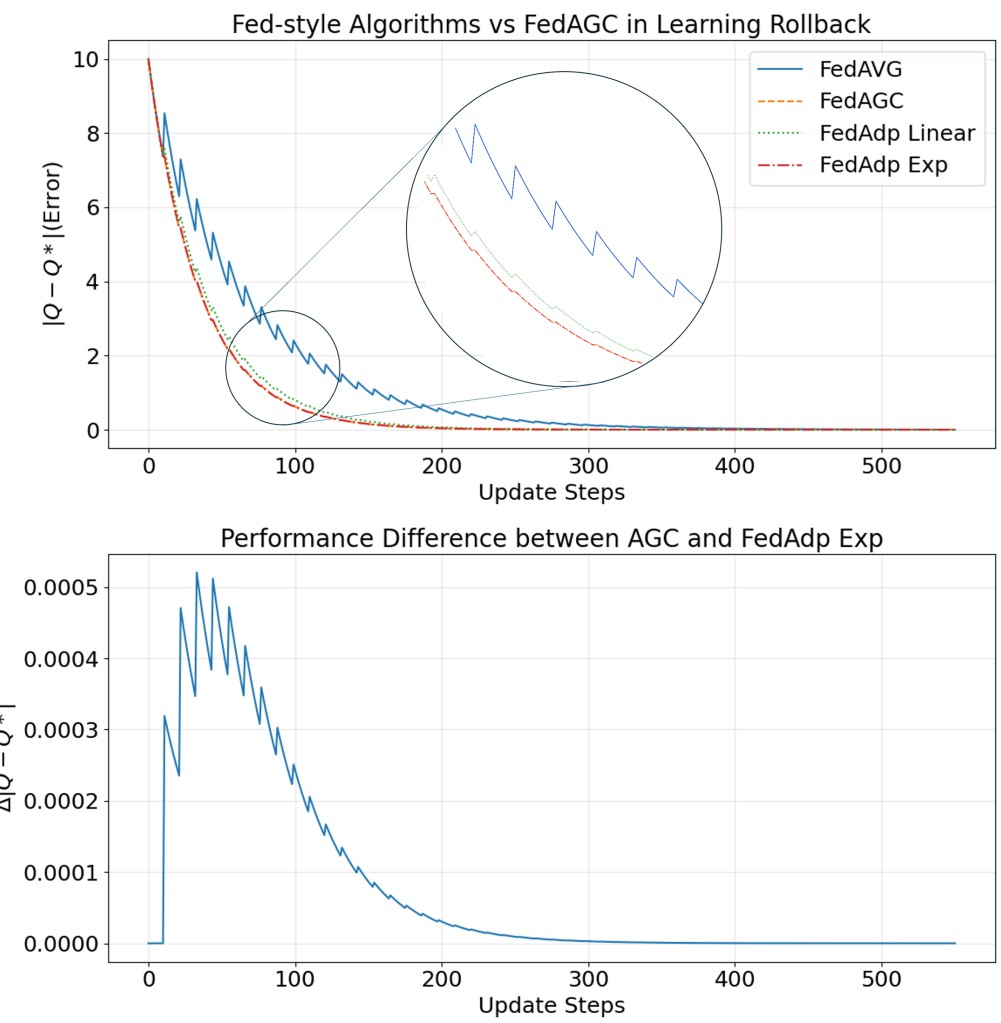

*Figure 10.* Learning performance comparison of our FedAGC with more Fed-style algorithm benchmarks in symbolic MDP.

### E.3. Details of Initialized States

In the cooperative navigation MPE experiments, agents are initialized at different fixed grid locations to induce heterogeneous uncertainty in their value estimates. The target position is fixed at $s_{\text{tar}} = (5, 5)$, and agent initial states are symmetrically placed around the grid boundary for different agent population sizes $M \in \{2, 4, 6, 8, 10, 12\}$.

Specifically, the initial positions of agents are defined as follows:

- $M = 2$:
$$\{(2, 2), (8, 8)\}$$

- $M = 4$:
$$\{(2, 2), (8, 2), (2, 8), (8, 8)\}$$

- $M = 6$:
$$\{(2, 2), (5, 2), (8, 2), (2, 8), (5, 8), (8, 8)\}$$

- $M = 8$:
$$\{(2, 2), (5, 2), (8, 2), (2, 5), (8, 5), (2, 8), (5, 8), (8, 8)\}$$

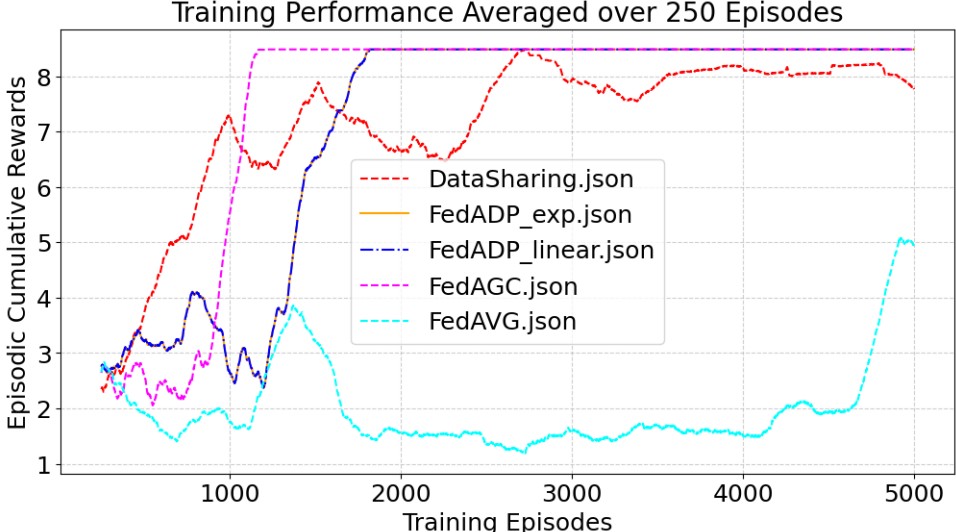

*Figure 11.* Training performance of FedAGC with comparison with more Fed-style benchmarks and direct DataSharing method in cooperative navigation task.

- $M = 10$:
$$\{(2,2),(5,2),(8,2),(2,4),(8,4),(2,6),(8,6),(2,8),(5,8),(8,8)\}$$

- $M = 12$:
$$\{(2,2),(4,2),(6,2),(8,2),(2,4),(8,4),(2,6),(8,6),(2,8),(4,8),(6,8),(8,8)\}$$

These configurations ensure that agents are initially distributed across the grid boundary, creating heterogeneous distances to the target and thus heterogeneous uncertainty in early value estimation.

For each configuration, the maximal achievable system score is computed as

$$\text{Score}_{\max} = \sum_{m=1}^{M} \left\| s_m^{(0)} - s_{\text{tar}} \right\|_2,$$

which corresponds to the case where all agents move directly toward the target without redundant actions. This value is used to normalize the training curves in the scalability experiments.

