# OpenReview forum: "Value Aggregation with Uncertainty in Online Decentralized MARL"
_ICML.cc/2026/Conference — ICML 2026 regular_

### Official Review · Reviewer_ZqYx · 2026-02-22

**Soundness:** 3
**Presentation:** 3
**Significance:** 3
**Originality:** 3
**Overall Recommendation:** 4
**Confidence:** 1

**Summary:**

This paper looks at how multiple AIs share what they’ve learned over a network. Usually, AIs share their "settings" (parameters) instead of raw data because it's more private and saves bandwidth. Most people just take the average of everyone’s settings, but the authors show this is a bad idea. They prove it causes a "learning rollback"—basically, an AI that has learned a lot can get its brain "reset" or messed up by averaging its data with another AI that hasn't learned much yet.

To fix this, they built a new tool called Adaptive Global Consensus (AGC). Instead of a simple average, AGC checks who actually knows what they’re talking about. It gives more "voting power" to the AIs with solid updates and less to the ones that are still guessing. They also designed a decentralized way for AIs to train and update these weights on their own. Finally, they proved mathematically that this makes the AIs learn faster and stay more stable, which they confirmed with real tests.

**Compliance With Llm Reviewing Policy:**

Affirmed.

**Final Justification:**

The rebuttal have addressed my concern and I will maintain my positive score.

**Key Questions For Authors:**

See weakness

**Limitations:**

Yes

**Strengths And Weaknesses:**

**Strength**

The paper does a great job explaining "learning rollback"—basically showing why just taking the average of everyone's data makes AIs learn slower or even get dumber. Their solution, AGC, is clever because it doesn't just average things; it uses smart math to push the learning forward faster based on who has practiced more.

**Weakness**
The math only works if you have a perfect, simple table of every possible move. It’s a huge question mark whether this would work in modern "Deep RL" where things are much more messy and complicated. Also, their "non-convex" weights (which can be weird numbers like negatives) might make the whole system crash or go haywire.

---

> ### Author Rebuttal · Authors · 2026-03-30
>
> We sincerely thank the reviewer's appreciation on our clear learning rollback issue formulation, clever design of AGC method to push learning forward faster and solid theoretical results. We will address the proposed question below.
>
> ---
>
> ### 1. Extensions to non-tabular form and Deep RL:
> This paper's position is set to first raise attention in the research community on the learning rollback issue caused by the difference between online learning in real training without the fundamental uniform ergodic assumption and offline learning. Then, to propose the universal theoretical learning method (Algorithm 1) to learn the proposed AGC weights (equation (13)) when the uncertainty can be estimated. As a basic RL algorithm for each individual Q-learning (TD(0)), whose uncertainty can be formally estimated, the construction of weights learning itself bridges the link between AGC weights computation and uncertainty level to resolve the learning rollback issue.
>
>
> This theoretical analysis and algorithm design can become a starting point for future research in the topic of real online learning when the uniform assumption no longer holds, and can still be applicable for complex scenarios, such as deep RL, and continuous state-action space, when their uncertainty can be estimated in a different framework for the whole state-action space. However, this is outside the scope of this manuscript and can be a standalone work for future extension.
>
>
>
> ### 2. Algorithm stability:
> It is true that a system with negative aggregation weights may crash. This is also our primary concern, and motivates us to derive robustness guarantees both in theory and experiment. In this paper, we have theoretically shown that our designed AGC weights are bounded, e.g., $||W_m^t||_{\infty}< c,\forall m\in[M],t\in[T]$, which is upper bounded by a constant independent to the learning parameters (Lemma 5.2 in page 5). Due to this upper-bounded property, the estimated Q-value variance is also bounded and controllable in theory. From the empirical aspect, we have also shown the proposed Algorithm 1 is robust to different noise levels (Fig. 6), and run the stress test when the noise signal arrived 60% of the reward signal, which still shows convergence.
>
> ----------
> We hope this clarification provides additional details to answer the reviewer's questions and removes any remaining doubts.

---

> > ### Author Rebuttal · Reviewer_ZqYx · 2026-04-02
> >
> > Thanks the authors for the rebuttal. I will keep my positive score.

---

### Official Review · Reviewer_FMFb · 2026-03-07

**Soundness:** 3
**Presentation:** 4
**Significance:** 3
**Originality:** 3
**Overall Recommendation:** 4
**Confidence:** 2

**Summary:**

The paper studies online decentralized MARL and the problem of aggregating Q-values from agents with different uncertainties. Standard convex aggregation can cause learning rollback. The authors propose **Adaptive Global Consensus (AGC)**, which weights agents by uncertainty, and a recursive **DTDE** update. They provide theoretical guarantees on convergence and variance, and show empirical improvements in dec-MARL tasks.

**Compliance With Llm Reviewing Policy:**

Affirmed.

**Final Justification:**

The authors fully address my concerns, therefore I decide to maintain my score.

**Key Questions For Authors:**

Are the baselines sufficient? The experiments seem to only compare against the basic FedAVG algorithm.

**Limitations:**

yes

**Strengths And Weaknesses:**

**Strengths:**

1. The paper explicitly analyzes how traditional convex aggregation can cause learning rollback under heterogeneous uncertainty and online updates, and proposes the AGC method with both theoretical and empirical support.

2. AGC dynamically adjusts aggregation weights based on each agent’s uncertainty, offering an intuitive approach that can be implemented fully decentralized while maintaining convergence and stability.

3. The paper provides theoretical analysis of AGC’s convergence and variance bounds, and validates its effectiveness through both MDP and MPE tasks, demonstrating accelerated convergence, rollback mitigation, and robustness.



**Weaknesses:**

1. Inconsistency in the full name of AGC. Line 46 writes *adaptive consensus mechanism (AGC)*, which should be corrected to adaptive global consensus mechanism (AGC). Line 157 writes *Adaptive Globe Consensus*, which should be corrected to Adaptive Global Consensus (the word “Globe” is a spelling error).
2. In Section 4.2,  Pow(D) measures the uncertainty related to an agent’s local updates. Could you clarify why Pow(D) accurately reflects uncertainty, and whether it corresponds to a formal statistical measure or is just a heuristic based on update counts?
3. Experiments are conducted on a symbolic MDP and a relatively simple MPE environment. There is no evaluation on larger-scale, high-dimensional, or more realistic MARL scenarios, which limits confidence in the method’s generalizability.
4. The weight computation requires collecting neighbors’ update counts and maintaining matrices like D. For large-scale networks, the computational overhead may be significant, but this is not analyzed.

---

> ### Author Rebuttal · Authors · 2026-03-30
>
> We sincerely thank the reviewer's high remark on this paper's clear motivation, learning rollback issue formulation, and convergence and stability analysis to the research gap in MARL both theoretically and empirically. We will address the questions below.
>
> ### 1. Minors:
> Thanks for the reviewer proposing some typos, we will correct them in the final version:
> - Line 46, _"adaptive consensus mechanism (AGC)"_ &rarr; ''adaptive global consensus mechanism (AGC)"; Line 157 _"Adaptive Globe Consensus"_ &rarr; "Adaptive Global Consensus"
>
> ### 2. Pow(D) and uncertainty:
>
> In the context of online policy evaluation, due to Bellman Contraction property for TD(0)-style Q-value updates (see Lemma 4.1 on page 4), the upper bounds of the estimated Q-values and the target Q-value can be characterized as power of $\gamma$ to the number of visitations.
> - We use such $\gamma$-powers to represent uncertainty (bias) is online policy evaluation. Since TD(0)-style Q-value updates is a fundamental updating dynamics in RL, most modern RL variants can still apply these $\gamma$-powers to evaluate the algorithm uncertainty [1]. This is a result built on learning theory for analyzing RL convergency and sample complexity [2] (it is a formal statistical measure).
> - Pow(D) is the matrix representation to compute such Q-values learning gap (uncertainty) in the whole state-action space, where Pow(D)(s,a) = $\gamma^{D(s,a)}$, $D(s,a)$ is the state-action visitation number updates between 2 aggregations at specific (s,a) pair. Since the power of $\gamma$ to the number of visitations is the formal statistical measure for uncertainty for the local updates, Pow(D) reflects the confidence gain between 2 aggregation. Our AGC constructs the weight based on this confidence gain (useful info) to guarantee convergence from continuous bias (uncertainty) decrease when aggregating.
>
> ### 3. Design of experiments:
>
> The experiments design in the main body follows the logic to correspond to the theoretical contribution of this theoretical paper. Section 6.1 confirms the existence of the learning rollback issue when doing convex aggregation even in a symbolic MDP. Section 6.2 and 6.3 empirically verifies the scalability, convergence acceleration and robustness of the proposed Algorithm 1, which corroborates our main theoretical performance analysis in Theorem 5.6. Up to that point, the paper is complete in showing the effectiveness of our Algorithm 1 both theoretically and empirically.
>
> The comparison benchmark FedAVG is a representative algorithm of convex Fed-learning that using simple averaging. For other carefully designed convex parameter aggregation methods, you will still experience similar issue in theory as shown in the motivation Fig. 1, and the only difference is you are expecting a smaller learning rollback, but can not bypass. The root of this issue is the convex weights design in online learning without uniform ergodic assumption as formally analyzed and motivated in Section 2.2.
>
> Therefore, we think the experiments demonstrated in the main body is sufficient to empirically prove the theorem, and we are open to include similar experiment results of comparing with other latest convex approaches (e.g. FedProx) in appendix.
>
> ### 4. Storage and communication complexity analysis:
> The exchanging parameters in Algorithm 1 are online Q-table $\hat{Q}_i$ and updating number of visitation table $D_i^t$ (both size |S||A|) for each connected agent pairs $i\in [M]$. The info exchange scheme uses pair-connection mode, which allows the exchange of only one pair of parameters. Therefore, the communication complexity is $\mathcal O(SA)$ , which is the same level as other parameter sharing methods or Federated-style learning algorithms.
>
> Parameter memory $\mathcal P$ can be created with fixed-size initialization and is to store the above received parameters recursively: 2M parallel |S||A| matrix, which can be compressed as a multi-dimensional tensor when calculation. The storage complexity $\mathcal O(MSA)$ is linear wrt scaling up the system. This is also one of the core advances for Algorithm 1 that allows parallel computing that supports full-tensor computation in real large-scale engineering scenarios (Fig. 4 shows the scalability).
>
> Alternatively, to save the M-linear storage with non-parallel computation, we can also free the memory with storage complexity $\mathcal O(SA)$ and do the incremental update immediately when received the exchange parameters for any aggregation pair recursively in the sequential mode when M is very large but at the cost of longer time consuming. In real industrial scenario, we normally select option 1 (our parallel computation algorithm) to save training time.
>
> We will also include this analysis in the appendix in the final version.
>
> ---
> [1] Woo et al., *The blessing of heterogeneity in federated Q-learning: Linear speedup and beyond*, JMLR, 2025.
> [2] Jin et al., *Is Q-learning provably efficient?*, NeurIPS, 2018.

---

> > ### Author Rebuttal · Reviewer_FMFb · 2026-04-01
> >
> > Thanks for the rebuttal.
> >
> > Can the authors provide experimental results compared with more baselines? (My question 1)

---

> > > ### Author Response · Authors · 2026-04-06
> > >
> > > Thanks for the reviewer's acknowledgment.
> > >
> > > Although the authors think it is sufficient to show the existence of the learning rollback issue and the advances of our proposed AGC method to tackle this issue in the current version both theoretically and empirically, we have included the following 2 more sets of benchmark comparison experiments with other Fed-style baselines, and plan to attach them to the appendix in the final version.
> > >
> > > Here we are considering 2 more Fed-style benchmarks (FedADP_lin[1], FedADP_exp[2]) to show the existence of learning rollback issue and compare the advances of our proposed FedAGC method. Formally, the normalized convex weights for 'FedADP_lin' and 'FedADP_exp' are proportional to the number of visits and exponential of number of visits.
> > >
> > > FedADP_lin:
> > > $$
> > > Q(s, a) = \sum_{k=1}^{K} \frac{N_k(s, a)}{\sum_{j=1}^{K} N_j(s, a)} Q_k(s, a);
> > > $$
> > > FedADP_exp:
> > > $$
> > > Q(s, a) = \sum_{k=1}^{K} \frac{\exp(N_k(s, a))}{\sum_{j=1}^{K} \exp(N_j(s, a))} Q_k(s, a).
> > > $$
> > > These two methods are popular adaptive Fed-style learning weights w.r.t. number of visitations. The experiment results are in the link: <https://docs.google.com/document/d/e/2PACX-1vTq8me8Jz4BnNyXdMyr7k9gDnwvAQdRza6F5TZvyluwEv1naJ-aGE6HaQ_ExxVhIFX0tStfC8COAkw9/pub>.
> > >
> > > The first set of experiment shows that even adaptive Fed-learning will still face learning rollback issue in the online symbolic MDP setting, as can be seen from small rollbacks in the Zoomed-in view of the learning process. Compared to simple average weights, both FedADP_lin (green curve) and FedADP_exp (red curve) can accelerate the learning process, and FedADP_exp's performance tends to approach our proposed FedAGC (orange curve), but the learning rollback can not be avoided. Subfigure 3 shows the zoomed-in view of the advances of FedAGC over FedADP_exp (the error of FedADP_exp is always larger than FedAGC, their difference is always positive), though their performances look similar.
> > >
> > > The second set of experiment is a complementary version of experiment in Section D.2 (Fig. 9), where we are training 2 agents to complete a cooperative navigation task. For small local update frequency (we choose $E=5$), the weights for FedADP_exp (orange line) and FedADP_lin (blue dotted) are similar, and we can observe their training performance (orange and blue curve) are similar in the training plot. Although both FedADP_exp and FedADP_lin manage to converge to the global optima, their convergence rate is slowed down due to the learning rollback issue compared to our proposed FedAGC (pink curve) method.
> > >
> > > I hope these additional baseline comparison experiments can thoroughly address the reviewer's questions and provided sufficient details to appreciate the importance of the issue and its impact in this setting.
> > >
> > > ---
> > > [1] McMahan B, Moore E, Ramage D, et al. Communication-efficient learning of deep networks from decentralized data[C]//Artificial intelligence and statistics. Pmlr, 2017: 1273-1282.
> > >
> > > [2] Wu H, Wang P. Fast-convergent federated learning with adaptive weighting[J]. IEEE Transactions on Cognitive Communications and Networking, 2021, 7(4): 1078-1088.

---

### Official Review · Reviewer_FuDG · 2026-03-09

**Soundness:** 3
**Presentation:** 3
**Significance:** 3
**Originality:** 3
**Overall Recommendation:** 3
**Confidence:** 3

**Summary:**

This paper provides a novel aggregation method for decentralized Multi-Agent Reinforcement Learning. The agents synchronize periodically and contribute to the global Q-table based on their confidence (visitation) in specific $(s,a)$ pairs, which makes the update of the $Q$-table more stable and accurate. The empirical studies also verify its utility.

**Compliance With Llm Reviewing Policy:**

Affirmed.

**Final Justification:**

During the rebuttal, one of the questions was not resolved; therefore, I maintain my score.

**Key Questions For Authors:**

See Weaknesses. I can increase to 4 if they are addressed.

**Limitations:**

yes

**Strengths And Weaknesses:**

Strengths:

The drawback of naive FedAvg is obvious, and this paper addresses it by proposing a weighted aggregation method. It also manages to establish the theoretical foundation on when, why and how their aggregation method works.

Weaknesses:

1. Figure 1 is confusing because it says the polar angle between the vector and the negative $x$-axis measures the confidence level. I think $0^\circ$ for the negative $x$-axis should be pointing to the left, and thus for certain? And, it says $||z_0||$ is the maximal (optimistic) initialization for the true action-value function. Why do you think the maximal initialization is optimistic?

2. Moreover, the uncertainty level is determined by the visitation frequency in the paper. And two agents have $2$ estimates, $z_1$ and $z_2$, the total number of visits is the sum of these two, then by averaging them, why does the uncertainty level go up as shown in Figure 1? In principle, the average of two estimates of a random variable has lower variance.

3. The dec-MARL setting is very similar to Federated Reinforcement Learning. Could the authors expand the related work section to include a comparison between these two settings, and maybe add some related work? I list some work that I think is relevant: [1], [2]

Minors: line 47, right side: in (1), I guess it should be "adaptive global consensus"? In (2), you didn't define the abbreviation "DTDE ".

[1] Woo et al., The Blessing of Heterogeneity in Federated Q-Learning: Linear Speedup and Beyond, Journal of Machine Learning Research, 2025

[2] Wang et al., On the convergence rates of federated Q-learning across heterogeneous environments, TMLR, 2025

[3] Wang et al., Federated TD Learning with Linear Function Approximation under Environmental Heterogeneity, TMLR, 2024

---

> ### Author Rebuttal · Authors · 2026-03-30
>
> We sincerely appreciate the reviewer recognizing this as a theoretically sound, well-presented, well-motivated, and significant work. We are also grateful for the questions, which we have thoroughly addressed below.
>
> ### 1. Figure 1 Explanation:
>    - The original figure description is accurate, and the polar angle is defined as the angle formed by the vector and the negative x-axis (pointing right, as defined in the illustrative figure). Here, the negative x-axis is **not** the conventional right-hand xy coordinate pointing to the left, but as in the illustrative figure, it is defined to the right, since the **x-axis arrow points to the left**, which defines the positive direction. If the reviewer has further suggestions to minimize confusion, we are happy to modify the figure accordingly. We hope our explanation now makes it clearer.
>    - The introduction of the x, y-axis here is only for the indication of axes and has no functional role in reality, since we are working in a polar system where only the polar angle and vector norm are of interest.
>    - Therefore, based on such a polar system, polar angle = 0 → vector at negative x-axis (pointing to the right) → defined as high uncertainty.
>    - **Optimistic initialization**: It is a standard RL concept (e.g., [1,2]), referring to initializing value estimates at their **maximum possible value within a given range**. It does not imply that the estimate is accurate or “optimistic” in a semantic sense, but rather follows a conventional initialization strategy.
>
> ### 2. Uncertainty Explanation:
>
> The uncertainty we consider here includes both bias and variance, not just variance alone. You are absolutely correct on pointing out that the summation over $N$ random variables results in reduced variance by $N$ **if their expectation and variance level are the same** (same distribution). However, neither the expectation nor the variance of the estimated parameters (Q-values) can be guaranteed to be the same at **any** given $(s,a)$ when they are merging among agents in online learning. This is why traditional convex federated learning fails in real online training, because the fundamental assumption (uniform ergodic) that convex Fed-style algorithms built on can not be guaranteed in real.
>
> - Initialized vectors are at both high bias and variance (pointing the right negative x-axis). Suppose we have 2 vectors $z_1$ and $z_2$ as in Fig. 1, where the polar angle of $z_1$ is smaller than $z_2$, which means $z_1$ is more uncertain (higher bias) than $z_2$. The result of a convex aggregation, its vector ending point lies between $z_1$ and $z_2$. Although you can see a reduced bias for the uncertain vector $z_1$, it is unfair for the certain $z_2$ as it increases the uncertainty level. **The overall uncertainty level did not go up for both estimates** in convex aggregation, and it's **unfair for the confident estimate.** This non-descending uncertainty property affects the convergence in online learning.
>
>
> - An analogy to this motivation in a numerical example, suppose we start to estimate a value from its maximal value, say 10. The target value is 1. One confident estimate is 3, and one uncertain estimate is 8. A convex aggregation will not reduce the uncertainty (bias) for both estimates, since it will lie between $[3,8]$. And our goal is to design a algorithm that drives the aggregation beyond confident estimate 3 to the true target value 1 ($z_t$ in Fig.1).
>
> ### 3. Literature on Federated Learning:
> We are grateful the reviewer brings some of the most recent research on this topic to our attention. In summary, the relation between networked MARL and Fed-learning, Fed-learning is one way to do parameter sharing (the other is experience sharing) in networked MARL [3], and heterogeneous Fed-learning considers real constraints in online learning.
>
> A detailed literature review is out of scope of this paper, and we are happy to include the extended literature review in our final version: <https://docs.google.com/document/d/e/2PACX-1vS34f_JjGXL1QAOEkDtN7fk4Z94e3fouOwOuMIZSu9whHoi5q6FHhwJzrX-O8bq_UxVuBbyx8X-O7Ag/pub>.
>
>
> ### 4. Minors:
> We appreciate the reviewer pointed the 2 minors. We will correct the first to 'adaptive global consensus mechanism (AGC)' to unify the term and define DTDE when it first appears in Line 50 right-column (instead of in Section 2.1 line 98).
>
> ### Summary:
> We hope this clarification resolves the concerns. Overall, we are grateful that the reviewer can consider to increase the score and we hope that the rigor of our theoretical contribution and the answer to the reviewers' questions have convinced the reviewer on the quality and importance of our work.
>
> ---
> [1] Woo et al., *The blessing of heterogeneity in federated Q-learning: Linear speedup and beyond*, JMLR, 2025.
> [2] Jin et al., *Is Q-learning provably efficient?*, NeurIPS, 2018.
> [3] Albrecht et al., *Multi-agent reinforcement learning: Foundations and modern approaches*, MIT Press, 2024.

---

> > ### Author Rebuttal · Reviewer_FuDG · 2026-04-01
> >
> > I have a follow-up question for "the summation over $N$ random variables results in reduced variance by $N$ if their expectation and variance level are the same (same distribution)". However, many concentration inequalities, for example, Hoeffding's Inequality, only require the random variables to be independent, not necessarily identical.

---

> > > ### Author Response · Authors · 2026-04-05
> > >
> > > Once again, we thank the reviewer for further remarks on using concentration law (e.g., Hoeffding's Law) to quantify and bound the aggregation uncertainty. However, we want to reiterate that the uncertainty we are considering here involves both bias and variance.
> > >
> > > Let us provide additional technical details to explain out point with more clarity. Let us recall that the concentration law indicates (taking Hoeffding's Law as an example) that, with high probability, the sum of bounded independent random variables will not deviate from its expected value by a certain value. In time series statistics, if $X_{i,t},i\in[1,M]$ are M independent random variables with bounds $X_{i,t}\in[a_{i,t},b_{i,t}]$, then the sum of these random variables $S_{M,t} = \sum_{i=1}^M X_{i,t}$ satisfies:
> > > $$
> > > P\left(\frac{1}{M}(S_{M,t}-E[S_{M,t}]) \ge \epsilon\right) \le \exp\left(-\dfrac{2M^2\epsilon^2}{\sum_{i=1}^M(b_{i,t}-a_{i,t})^2}\right), \forall \epsilon\ge 0.
> > > $$
> > >
> > > In our online dec-MARL sharing Q-value setting, although the estimated Q-values $X_{i,t} = \hat{Q}\_{i,t}(s,a), i\in[1,M]$ are random variables bounded within $[0,H/(1-\gamma)]$, the $E[S_{M,t}]$ is still a random variable w.r.t. timestep $t$ (see Theorem 5.4 in the main body), and this random variable has no uniform convergence relation (on $t$) to its optima $Q^{\*}(s,a)$.  The bias here is defined as $E[\frac{S_{M,t}}{M}] - Q^{\*}$. Therefore, the concentration does not provide any uncertainty bound properties to the optima $Q^*(s,a)$, which, in turn, causes traditional Fed-style convex aggregation non-convergence or convergence to some sub-optima.
> > >
> > > Another way to describe this problem is that we are aggregating random variables $X_{i,n_i}$, where each $X_{i,n_i}$ forms a decreasing sequence that converges to $X^{\*}$ as $n_i\rightarrow\infty$, starting from maximal initialization. Here, $n_i = N_i(s,a)$ denotes the number of visits to the specific $(s,a)$. Without uniform ergodic assumption, although we have the constraint $\sum_{(s,a)}\sum_{i=1}^M n_i= Mt$, we do not know how $n_i$ is distributed on the whole state-action space. Therefore, we have no guarantee for the expectation $E[\frac{1}{M}\sum_{i}^M X_{i,n_i}]$ convergence to the optima $X^*$  as $t$ increases in general.
> > >
> > >
> > > As a simple case, consider when the policy entails to execute purely no-operation for all agents except a well-trained agent M, and agents only collect sample on the initialized $(s,a)$ $t$ times while other states are not updated. Then, there exists a $(s,a)$ that all agents never collected data, except agent M. So $X_{i,t} = \hat{Q}\_{i,t}(s,a) = H/(1-\gamma), \forall i\in[1,M-1]$, and $X_{M,t} = \hat{Q}\_{M,t}(s,a) = Q^{\*}$. The expectation $E[\frac{1}{M}S_{M,t}]\approx E[X_{1,t}] = H/(1-\gamma)$ when M is large. In this scenario, Hoeffding's concentration law provides the following, when $t\rightarrow \infty$,
> > > $$
> > > P\left(\frac{S_{M,t}}{M}-H/(1-\gamma) \ge \epsilon\right) \le \mathcal O\left(\exp(\frac{M\epsilon^2}{H^2})\right), \forall \epsilon\ge 0.
> > > $$
> > > For the well-trained agent M, the baseline expectation has been increased from a certain value $Q^{\*}$ to a very uncertain value $E[X_{1,t}]$ and is not related to the $Q^{\*}$ even when $t\rightarrow\infty$. This global aggregation is not only sending to all uncertain agent $1: M-1$, but also the well-trained agent M, that avoids the learning proceeding. Therefore, this concentration provides no useful information to bound the well-trained agent M's uncertainty towards the optima, since no guarantee of the expectation decreasing to the optima.
> > >
> > > Alternatively, our designed AGC method provides the monotone $n_i$ updates scheme and ensures that $E[\frac{S_{M,t}}{M}]$ decreases towards (i.e., converges to) the optima, which guarantees the AGC aggregation converges in online setting without uniform ergodic assumption.

---

### Official Review · Reviewer_L7c1 · 2026-03-13

**Soundness:** 3
**Presentation:** 2
**Significance:** 3
**Originality:** 3
**Overall Recommendation:** 4
**Confidence:** 4

**Summary:**

This paper characterize the learning rollback phenomenon in decentralized MARL, and proposes adaptive global consensus (AGC) mechanism for Q-value aggregation. The AGC is a count-based method for assigning state-action-pairwise aggregation weights (possibly negative) for Q-value aggregation, thus avoiding the under-trained agents to slow down the training of well-trained agents. The authors provide theoretical guarantees to the convergence and variance reduction properties of their proposed method.

**Compliance With Llm Reviewing Policy:**

Affirmed.

**Final Justification:**

During the rebuttal, the authors demonstrated that the learning rollback phenomenon is fundamental to online decentralized learning without the uniform ergodic assumption, and provide a clear explanation on their experimental design. My concerns have been fully addressed. Therefore, I raise my score.

**Key Questions For Authors:**

See Weaknesses.

**Limitations:**

yes

**Strengths And Weaknesses:**

Strengths:
1. The proposed method is well-motivated.
2. The authors provide a solid theoretical foundation to their proposed method.

Weaknesses:
My primal concern is the learning rollback phenomenon itself. The authors did not provide adequate evidence to the existence of this phenomenon in realistic scenarios. Intuitively, if each agent have more exploration between aggregation, or if they use policy evaluation methods with higher sample efficiency instead of TD(0), or if the state-action space is not tabular but has some inner structure so as for generalization, this phenomenon may not exist.

---

> ### Author Rebuttal · Authors · 2026-03-30
>
> We thank the reviewer for recognizing that our work is *well-motivated* and supported by a *solid theoretical foundation*. We address the concerns below.
>
> ### **On the existence of the rollback phenomenon**
>
> The reviewer suggests that larger exploration, sample-efficient RL, or function approximation **may** remove the phenomenon. We clarify that the issue is **fundamental to online dec-learning without the uniform ergodic assumption**, rather than tied to a specific algorithm. We note that this concern is phrased as a possibility, which appears inconsistent with reviewer's confidence. We eliminate such possibility with the analysis below.
>
>
> In this regime, agents exhibit heterogeneous uncertainty (bias and variance) over the state-action space **due to non-uniform visitation**. This is a general property of online learning [1] and is supported both theoretically (Section 3) and empirically (Section 6.1, Fig. 3).
>
> The learning rollback phenomenon was also **observed in our real-world drone swarm experiments** for cooperative learning in decentralized MARL systems. In such settings, parameter sharing via convex aggregation (e.g., FedAvg-style updates [4]) leads to **suboptimal convergence in online regimes**. This motivates the formulation in Section 3, demonstrating that the phenomenon is not merely theoretical but arises in realistic systems.
>
> - **Sample-efficient RL does not eliminate rollback:**
>
> Many federated/offline RL analysis [2-4] rely on the *uniform ergodic assumption* to ensure uniform convergence across the state-action space. This assumption rarely holds in online settings. As a result, even highly sample-efficient algorithms cannot prevent **heterogeneous uncertainty during training**, unless all agents are already fully trained on the whole state-action space—at which point exchanging info (further learning) becomes unnecessary.
>
> The issue is particularly pronounced when the state-action space is large (e.g., $|S||A| \gg T$), where estimates are inherently high-bias for local updates due to limited samples. Convex aggregation of such estimates does not reduce bias, which directly motivates our approach. The core challenge is thus how to mitigate uncertainty (both bias and variance) while agents are simultaneously learning and communicating—this is also a central question in heterogeneous federated learning [2–5].
>
> - **Increasing exploration between aggregations:**
>
> This corresponds to the parameter $E$ in our Algorithm 1.  While larger $E$ improves local estimation, it **does not remove heterogeneous uncertainty across the full state-action space**. Both our theory and experiments show that rollback persists for varying $E$.
>
> Moreover, selecting $E$ introduces a **tradeoff** to maximize the sample efficiency: larger $E$ improves parallel local accuracy for each agent, while smaller $E$ enjoys more frequent aggregation advances. We agree this is an interesting direction for future work.
>
> - **Function approximation / non-tabular settings:**
>
> The learning rollback phenomenon is **not limited to tabular settings**. In continuous or structured state-action spaces, non-uniform online data collection still induces heterogeneous uncertainty across regions of the space.
>
> Therefore, the phenomenon persists in non-tabular settings as well. Our framework remains applicable as long as uncertainty can be estimated, and the key issue—aggregation over heterogeneous uncertainty—remains unchanged.
>
> ### **On scores and consistency**
>
> We respectfully note an **inconsistency** between the qualitative assessment and the numerical scores. While the work is acknowledged as well-motivated with solid theory, it receives **low scores on originality (1) and significance (1)**. We would appreciate further clarification on this point, as theoretical grounding and problem formulation are central aspects of both originality and impact.
>
> ### **Summary**
>
> We emphasize that learning rollback is a **general phenomenon in online decentralized learning without uniform ergodicity**, rather than an artifact of specific algorithms or tabular settings. We provide both empirical evidence and theoretical justification supporting its existence and relevance.
>
> We hope this clarification addresses the reviewer’s concerns and respectfully request reconsideration of the scores.
>
> ---
>
> [1] Albrecht, S. V., Christianos, F., & Schäfer, L. *Multi-agent reinforcement learning: Foundations and modern approaches*. MIT Press, 2024.
> [2] Wang, M., Yang, P., & Su, L. “On the convergence rates of federated Q-learning across heterogeneous environments.” *arXiv:2409.03897*, 2024.
> [3] Woo, J., Joshi, G., & Chi, Y. “The blessing of heterogeneity in federated Q-learning: Linear speedup and beyond.” *JMLR*, 26(26), 2025.
> [4] Li, X., et al. “On the convergence of FedAvg on non-IID data.” *arXiv:1907.02189*, 2019.
> [5] Reddi, S., et al. “Adaptive federated optimization.” *arXiv:2003.00295*, 2020.

---

> > ### Author Rebuttal · Reviewer_L7c1 · 2026-04-03
> >
> > Thank you for the explanation on the existence of the learning rollback phenomenon. This largely address my concern as the issue is fundamental to online decentralized learning without the uniform ergodic assumption. However, this raises questions on the experimental design in Section 6. In the experiments, the heterogeneous across agents are solely resulted from different update frequencies (In Fig.3(left), no learning rollback for $E_1=10$ and $E_2=9$), instead of non-uniform visitation as the authors claim here and in Section 1 of the paper. Can the authors comment on this?
> >
> > I believe that clearly demonstrating the existence and the cause of the learning rollback phenomenon is essential to this paper, and I will adjust the scores accordingly. The previous low scores on originality and significance were based on the questionable existence of the learning rollback phenomenon, and is sure to be adjusted.

---

> > > ### Author Response · Authors · 2026-04-05
> > >
> > > Thanks for the reviewer proposing further questions, and the willingness raise the score.
> > >
> > > Section 6.1 provides the basic symbolic MDP experiment to show that the learning rollback issue actually exists. In online learning, non-uniform visitation over the whole state-action space is the cause of the observed learning rollback issue. Alternatively, "non-uniform visitation over the whole state-action space" is equivalent to say for any given $(s,a)$ pair, the number of visitation on the $(s,a)$ varies, thus causing the heterogeneous uncertainty for different agents. Think of a well-trained agent that has updated its parameters much more than an initialized agent without parameter updates, and the 2 agents have various uncertainty levels to the optimal parameter.
> > >
> > > To provide a direct demonstration of "non-uniform visitation", we simplify the whole state space to a fixed single state space, and design the 2 agents to make local updates with different frequencies, indicating different number of updates (heterogeneous uncertainty) when aggregating parameters. In summary, the first experiment is designed to investigate what will happen when convex-aggregating parameters with heterogeneous uncertainty, and the learning performance (rollback) is measured by the error distance.
> > >
> > > A detailed explanation discussing the experiment in detail follows. Two agents, one more confident with local update rate $E_1=10$ is communicating with one less confident agent with $E_2 \in\{0,2,5,9\}$. The more close for $E_2$ is to $E_1$, their local updates difference is smaller, and their parameter uncertainty heterogeneity at the fixed state is smaller. You can see the direct relation between the parameter uncertainty heterogeneity and learning rollback in Fig. 3 (left): larger heterogeneity ($E_2=0$) result in more severe rollback, smaller heterogeneity ($E_2=9$) result in lighter rollback. We clarify that in principle there is still a tiny rollback ($\gamma^{E_2=9}(1-\gamma)$) that is not visually observable in the figure and the learning rollback issue is not resolved in principle. This is also not indicating larger $E_2$ can eliminate the learning rollback issue, since we fixed $E_1=10$ here. You can only expect a more close $E_1$ and $E_2$ pair (more similar parameter uncertainty), a less severe learning rollback issue you can observe.
> > >
> > > We hope this further explanation has thoroughly addressed the reviewer's concern and provided sufficient details to appreciate the importance of the issue and its impact in this setting.

---

### Decision · Program_Chairs · 2026-04-30

**Decision:**

Accept (regular)

**Comment:**

This paper considers the problem of aggregate value estimation in MARL. The overall strength and weakness of the paper are as follows.

Strength
- The proposed method is well motivated.
- Theoretical results are provided to ground the method.

Weakness
- The use of terminology should be improve to avoid confusion.
- The experimental results should be strengthened and better discussed.

While there are concerns about the paper, most of them are well addressed during the rebuttal phase.